# A structural polymer for highly efficient all-day passive radiative cooling

Tong Wang[1], Yi Wu[1], Lan Shi [1], Xinhua Hu [1], Min Chen [1] & Limin Wu [1]✉

All-day passive radiative cooling has recently attracted tremendous interest by reflecting sunlight and radiating heat to the ultracold outer space. While some progress has been made, it still remains big challenge in fabricating highly efficient and low-cost radiative coolers for all-day and all-climates. Herein, we report a hierarchically structured polymethyl methacrylate (PMMA) film with a micropore array combined with random nanopores for highly efficient day- and nighttime passive radiative cooling. This hierarchically porous array PMMA film exhibits sufficiently high solar reflectance (0.95) and superior longwave infrared thermal emittance (0.98) and realizes subambient cooling of ~8.2 °C during the night and ~6.0 °C to ~8.9 °C during midday with an average cooling power of ~85 W/m$^2$ under solar intensity of ~900 W/m$^2$, and promisingly ~5.5 °C even under solar intensity of ~930 W/m$^2$ and relative humidity of ~64% in hot and moist climate. The micropores and nanopores in the polymer film play crucial roles in enhancing the solar reflectance and thermal emittance.

[1] Department of Materials Science and State Key Laboratory of Molecular Engineering of Polymers, Fudan University, 200433 Shanghai, China.
✉email: lmw@fudan.edu.cn

Although the climate-science community is attempting to find efficient solutions for the accelerating global warming and greenhouse gas emissions, few concrete actions have been taken to resolve climate change[1–3]. Conceptually, one of the most efficient strategies is to reduce the amount of solar irradiance absorbed by the Earth[4], e.g., through solar radiation management (SRM), to slow or reverse global warming[5–7]. The basic idea behind SRM is to seed reflective particles into the Earth's stratosphere to reduce solar absorption, which might cause potentially dangerous threats to the Earth's basic climate operations[8,9]. One possibly alternative approach is passive radiative cooling—a sky-facing surface on the Earth spontaneously cools by radiating heat to the ultracold outer space through the atmosphere's longwave infrared (LWIR) transparency window ($\lambda \sim 8$–$13\,\mu m$)[10–14]. However, passive daytime radiative cooling (PDRC) to a temperature below ambient under direct sunlight is a particular challenge because most of the naturally available thermal radiation materials also absorb incident solar irradiance and rapidly heat up under exposure to the Sun[15,16]. Accordingly, designing and fabricating efficient PDRC with sufficiently high solar reflectance ($\bar{\rho}_{solar}$) ($\lambda \sim 0.3$–$2.5\,\mu m$) to minimize solar heat gain and simultaneously strong LWIR thermal emittance ($\bar{\varepsilon}_{LWIR}$) to maximize radiative heat loss is highly desirable[17,18]. When the incoming radiative heat from the Sun is balanced by the outgoing radiative heat emission, the temperature of the Earth can reach its steady state[4,19]. Thus, PDRC technology is very promising to considerably decrease the use of compression-based cooling systems (e.g., air conditioners) and has a significant impact on global energy consumption[20–24].

The first theoretical design of a metal-dielectric photonic structure for PDRC was presented by Raman et al.[25] in 2013 by tailoring the material spectrum responses for continuous daytime radiative cooling. Then, they first experimentally achieved PDRC via a precision-designed nanophotonic radiative cooler in 2014[26]. This cooler, consisting of seven alternating dielectric layers deposited on top of a silver mirror, cooled to 4.9 °C below ambient temperature by reflecting 97% of incident sunlight while strongly and selectively emitting in the atmospheric transparency window. Nonetheless, many photonic structures suffer from a high manufacturing cost and large-scale production limits[23,26–28]. Another pioneering strategy was recently developed by Yin et al.[29], who created a glass-polymer hybrid metamaterial thin film consisting of silica ($SiO_2$) microspheres randomly distributed in the matrix material of polymethylpentene via the scalable-manufactured roll-to-roll polymer extrusion process. With the assistance of a silver coating, the metamaterial was able to exhibit >93% infrared emissivity and reflect ~96% of solar irradiance, achieving a noontime radiative cooling power of 93 W/m[2] under direct sunshine. The introduction of polymer-based radiative cooling materials can greatly improve the scalability and applicability of PDRC systems in practical applications[18,30–33]. Very recently, instead of using a reflective metallic mirror, state-of-the-art PDRC designs, such as porous polymer coatings[17,34,35], polymeric aerogels[36], white structural wood[37] and cooling paints[38,39], have attracted considerable attention because of their high cooling performance, simplicity, applicability and economical efficiency. For example, Yu et al.[17] made remarkable progress in the design of PDRC poly(vinylidene fluoride-co-hexafluoropropene) coatings with random micro-/nano-pores through a phase inversion-based method, demonstrating high solar reflectance (0.96 ± 0.03), as well as high longwave infrared emittance (0.97 ± 0.02) that enabled cooling up to ~6 °C and ~3 °C below ambient temperature under direct sunlight in dry southwestern USA and south Asia, respectively. Nonetheless, almost all of the PDRC prototypes reported so far performed well in an arid atmosphere (e.g., total precipitable water (TPW) <10 mm) and at relatively low environmental

temperatures (e.g., below 25 °C)[26,29,36,37], rather than in hot (e.g., 30 °C or above) and humid (e.g., TPW å 20 mm) regions[17,30]. Theoretically, atmosphere humidity or TPW heavily influences the absorbed power density of atmospheric radiation and the magnitude of the cooling performance. In a dry environment, the atmospheric window is open not only in the 1st atmospheric transparency window (8–13 μm) but also in the 2nd atmospheric transparency window (16–25 μm), making efficient thermal radiation possible[23,30]. However, in a hot and humid environment, the transmissivity of the atmosphere slightly decreases in the 1st atmospheric transparency window and dramatically drops in the 2nd atmospheric transparency window. Thus, the cooling performance naturally becomes lower and is even limited due to the increased downwelling atmospheric radiation induced by higher humidity and temperature[40–42]. Therefore, developing facile, scalable, and cost-effective PDRC for practical thermal radiation applications, including hot and humid regions, still remains a great challenge.

Inspired by these intelligent reports and above considerations, herein, we design and report a hierarchically porous array PMMA ($PMMA_{HPA}$) film with a close-packed micropore array on the surface combined with abundant random nanopores inside by a template method. The as-obtained $PMMA_{HPA}$ film demonstrates excellent $\bar{\rho}_{solar}$ (0.95) and $\bar{\varepsilon}_{LWIR}$ (0.98), with not only subambient cooling of as high as ~8.2 °C during the night and of ~6.0 °C to ~8.9 °C during midday with an average cooling power of ~85 W/m[2] under solar intensity of ~900 W/m[2], but also ~5.5 °C even under solar intensity of ~930 W/m[2] and relative humidity of ~64% in hot and moist subtropical marine monsoon climate, which was not reported previously. Both experimental evidence and theoretical calculations verify that both the dense micropore array on the surface and the random nanopores in the $PMMA_{HPA}$ film play crucial roles in enhancing the solar reflectance and thermal emittance, which provides deep insight into devising superb PDRC technologies.

## Results and discussion

$\bar{\rho}_{solar}$ and $\bar{\varepsilon}_{LWIR}$ of the PMMA$_{HPA}$ film. Fig. 1a illustrates the fabrication process of $PMMA_{HPA}$. In brief, a monolayer of a hexagonally close-packed 5 μm $SiO_2$ array is fabricated using a facile unidirectional rubbing assembly method[43,44], as shown in Fig. 1b, c. A dispersion of PMMA and 200 nm $SiO_2$ nanospheres in acetone is then infiltrated into the $SiO_2$ monolayer template. After the rapid evaporation of acetone in air, the obtained PMMA/$SiO_2$ composite displays randomly distributed $SiO_2$ nanospheres and regularly distributed $SiO_2$ microspheres, as indicated by the micrograph and EDS elemental mappings (Fig. 1d–g). After removal of the $SiO_2$ nanospheres and the monolayer template by etching in hydrofluoric acid aqueous solution, a hierarchically porous array PMMA ($PMMA_{HPA}$) film with ordered symmetrical micropores (~4.6 μm diameter) and randomized nanopores (~250 nm average diameter) can be obtained (Fig. 1h, i and Supplementary Figs. 1a–c). Fourier transform infrared (FTIR) spectra and thermogravimetric analysis (TGA) confirm that $PMMA_{HPA}$ is entirely composed of organic polymer without residual inorganic $SiO_2$ (Supplementary Figs. 1d and e), while the 49.4 wt% of $SiO_2$ before etching indicates a high porosity of $PMMA_{HPA}$ (Supplementary Fig. 1f).

Figure 2 demonstrates the spectral reflectance and emissivity of the $PMMA_{HPA}$ film with ~160 μm effective thickness according to the normalized ASTM G173 Global solar spectrum and the LWIR atmospheric transparency window. The $PMMA_{HPA}$ film with ~60% porosity presents a high average solar reflectance ($\bar{\rho}_{solar}$ = 0.95, Fig. 2a and Supplementary Fig. 2a), which ensures excellent

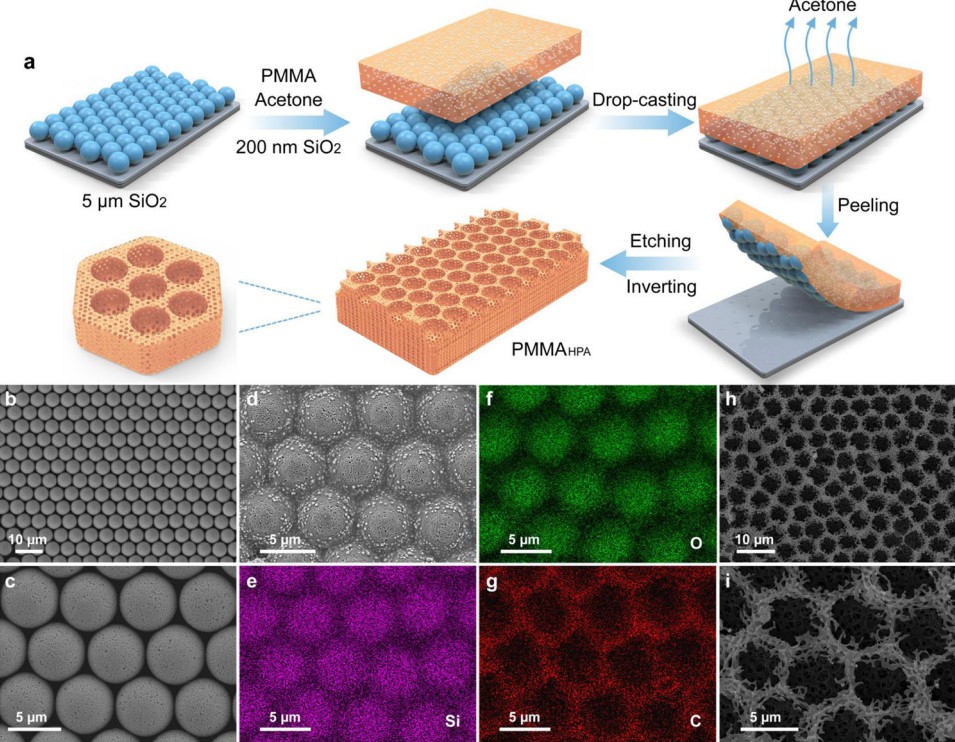

**Fig. 1 Fabrication and characterization of the PMMA$_{HPA}$ film. a** Schematic illustration of the fabrication of PMMA$_{HPA}$ with a hierarchically porous array. **b**, **c** SEM micrographs of hexagonally close-packed monolayer SiO$_2$ templates. **d** SEM micrograph of PMMA/SiO$_2$ composite. **e–g** EDS elemental mappings of Si, O, and C in **d**, showing the randomly distributed SiO$_2$ nanospheres and regularly distributed SiO$_2$ microspheres of the composite. **h**, **i** SEM micrographs of PMMA$_{HPA}$ showing an ordered symmetrical micropores array made of hierarchical randomized nanopores.

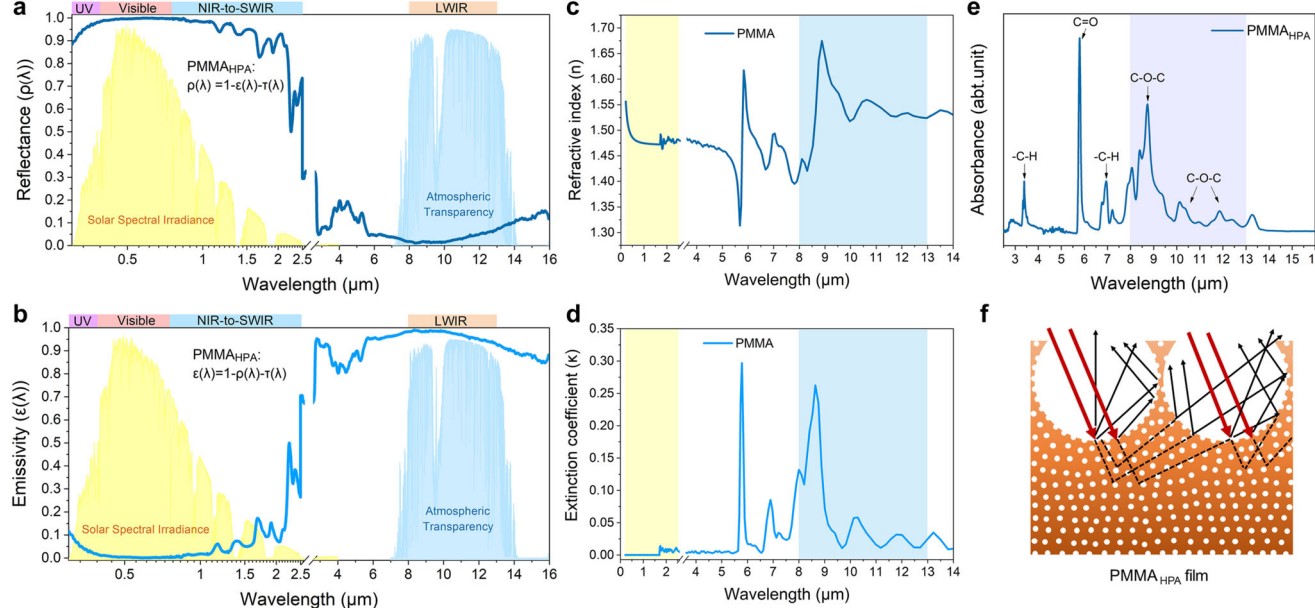

**Fig. 2 Spectroscopic response of the PMMA$_{HPA}$ film. a**, **b** Spectral reflectance and emissivity of the PMMA$_{HPA}$ film with ~160 μm effective thickness along with the normalized ASTM G173 Global solar spectrum and the LWIR atmospheric transparency window. **c**, **d** Spectral refractive index (n) and extinction coefficient (κ) of PMMA, showing negligible absorptivity in the solar range and multiple extinction peaks in the LWIR wavelengths. **e** Absorbance spectrum of PMMA$_{HPA}$ measured with ATR-FTIR spectroscopy. **f** Schematic diagram showing the periodic re-entrant structure with nano/microscale pores is conducive to enhance the total scattering efficiency by multiple reflections.

reflection of sunlight from all incidences and minimizes the solar heat gain. Meanwhile, the film also shows a high thermal emittance over a broad bandwidth in the mid-infrared and still emits a significant part of its thermal energy even at large emission angles ($\bar{\varepsilon}_{LWIR} = 0.98$, Fig. 2b and Supplementary Fig. 2b), which can enable strong emission of heat to the cold sink of outer space through the atmospheric transparency window at different angles in relation to the sky. One remarkable feature of our

structural polymer is that the periodic ordered ~4.6 μm micro-pores can scatter sunlight of ultraviolet-visible-near-infrared (UV-Vis-NIR) wavelengths, and the abundant, random ~250 nm nanopores greatly reduce the mean scattering path and transmission through the material, which further enhances the scattering of shorter visible wavelengths[45,46]. The combination of PMMA with air voids, one of which has a high refractive index and the other has a low refractive index ($\Delta n = n_{PMMA} - n_{air} = 1.49 - 1 = 0.49$), could provide a sharp refractive index transition across polymer-air boundaries and yield the efficient solar scattering and the required strong sky window absorptance without surface obstruction[34].

As one of the most widely used and low-cost polymers, pristine PMMA film has ideal intrinsic properties to enable high-performance PDRC applications[47]. Figure 2c, d show that PMMA has negligible extinction coefficient in the solar wavelengths and multiple extinction peaks at the 8, 8.6, 10.3, 11.8, and 13.2 μm within the LWIR window, which should result from the different vibrational modes of its molecular structure. These properties keep the heat gain from sunlight to a minimum and contribute to a large amount of infrared absorption/emission in the atmospheric transparency window, which is responsible for the superior PDRC. Furthermore, the absorbance spectrum measured with attenuated total reflectance-Fourier transform infrared spectroscopy (ATR-FTIR) exhibits strong infrared absorption due to C–O–C stretching vibrations between 770 and 1250 cm$^{-1}$ (8–13 μm, Fig. 2e), which coincidently lie in the atmospheric transparency window. Importantly, the periodic re-entrant structure with hierarchical nano/microscale pores is conducive to enhance the total scattering efficiency and increase the probability of infrared absorption/emission through multiple diffuse reflection at various incident angles (Fig. 2f). All these features benefit the radiative heat exchange between the cooling structural polymer and the atmosphere, causing sufficiently high $\bar{\rho}_{solar}$ and $\bar{\varepsilon}_{LWIR}$ of our PMMA$_{HPA}$ film to achieve all-day passive radiative cooling.

Different from transparent pristine PMMA, the PMMA/SiO$_2$ composite film is translucent due to the absorptance in the UV and visible regions, and the corresponding PMMA$_{HPA}$ film has negligible transmittance because its plenty of pores can efficiently scatter sunlight of all wavelengths (Supplementary Figs. 3a–d). The spectral transmittance of the PMMA$_{HPA}$ film decreases with increasing thickness (Supplementary Fig. 4). Accordingly, the solar reflectance appears to have a more pronounced increasing trend with thickness than the thermal emittance in the range of 8–13 μm (Fig. 3a–c), which likely arises from the increased backscattering of light from the thicker, nonabsorptive, porous PMMA layer. Evidently, even the emittance of the only effectively ~2 μm thick PMMA$_{HPA}$ film can reach 0.85. This suggests that the porous structure is sufficient to augment the intrinsic emittance of PMMA$_{HPA}$. Further, we experimentally and theoretically demonstrated the influence of pore sizes and porosity on the optical performance. As clearly seen in Supplementary Fig. 5a, the PMMA$_{HPA}$ with ~500 nm nanopores shows a high-level $\bar{\rho}_{solar}$, but its scattering of short wavelength region is inferior to that of ~200 nm nanopores. The reflectance of PMMA$_{HPA}$ with ~100 nm nanopores drops gradually in the visible and NIR-to-SWIR ranges (0.4–2.5 μm), leading to a lower $\bar{\rho}_{solar}$. Such results coincide with the finite-difference time-domain (FDTD) simulation data (Supplementary Fig. 6a), indicating nanopores ranging in diameter from 200 to 300 nm are optimum to reinforce the scattering of UV and visible wavelength region. For the optimization of micropore sizes, we established a model of ordered monolayer micropore with different sizes in the range of 1 to 10 μm. Given the simulation results, one could derive that micropores ranging from 5 to 7 μm in diameter would highly contribute to the solar reflectance (Supplementary Fig. 6b). Thus, combining the optimized nanopore size of 200–300 nm and micropore size of 5–7 μm, the maximum $\bar{\rho}_{solar}$ of our hierarchical porous PMMA film can be obtained (Supplementary Fig. 6c). Moreover, the solar reflectance of the PMMA$_{HPA}$ film increases with increasing porosity, while its thermal emittance does not

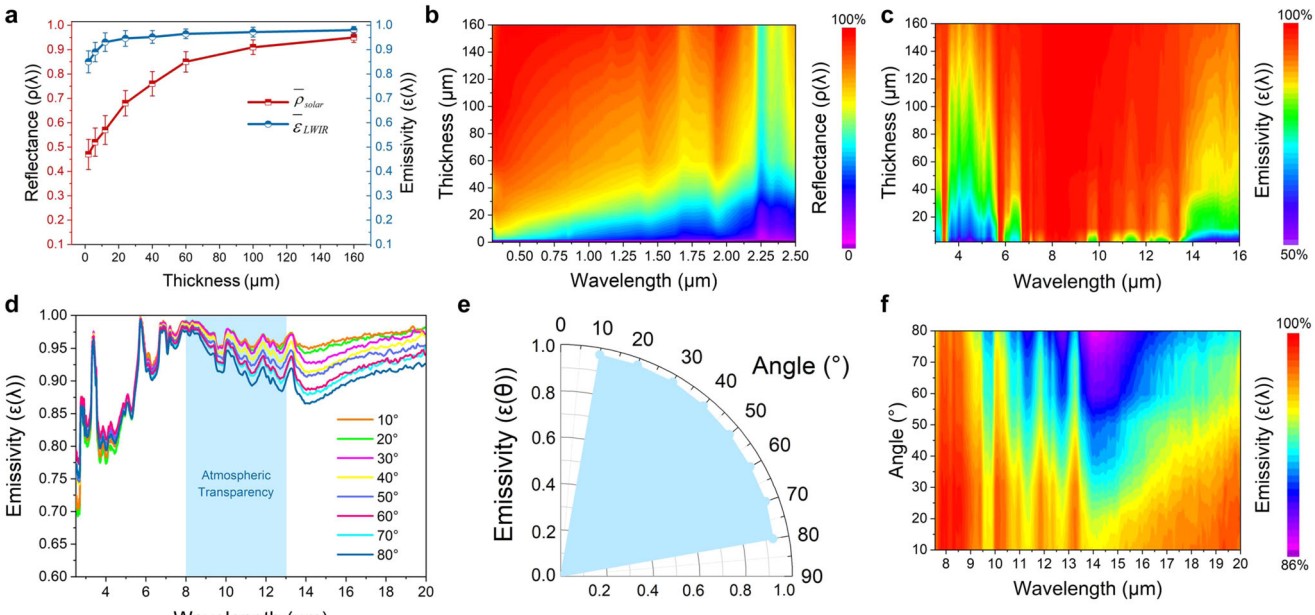

**Fig. 3 Variation in $\bar{\rho}_{solar}$ and $\bar{\varepsilon}_{LWIR}$ of the PMMA$_{HPA}$ films with effective thickness and polarization angle. a** Variation in $\bar{\rho}_{solar}$ and $\bar{\varepsilon}_{LWIR}$ of the PMMA$_{HPA}$ films with effective thickness. The error bars represent the standard deviation. **b, c** Measured reflectance and emissivity spectra of the PMMA$_{HPA}$ films as a function of the film effective thickness. **d** Infrared emissivity spectra of a PMMA$_{HPA}$ film at different polarization angles (θ) from 10° to 80°. **e** Polar distribution of the average emissivity across the atmospheric window of the PMMA$_{HPA}$ film at different polarization angles (θ) from 10° to 80°. **f** Measured polarization-dependent infrared emissivity spectra of the PMMA$_{HPA}$ films.

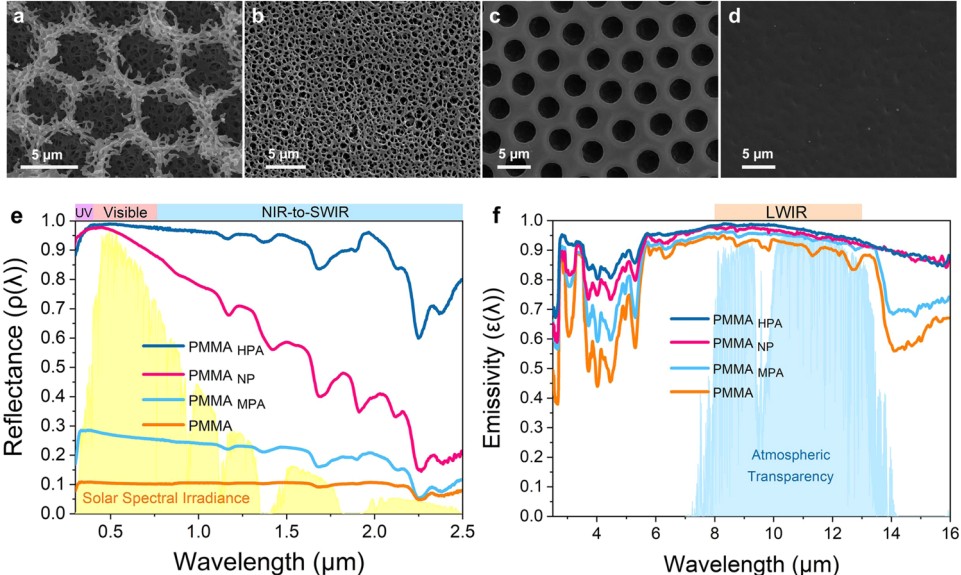

**Fig. 4 SEM micrographs and optical properties of different types of PMMA films. a–d** SEM micrographs of the PMMA$_{HPA}$, PMMA$_{NP}$, PMMA$_{MPA}$, and pristine PMMA films. **e** Reflectance spectra across the solar wavelengths of different types of PMMA films. **f** Infrared emissivity spectra of different types of PMMA films.

significantly change (Supplementary Fig. 5b). FDTD simulation results also show that the nanopore density or porosity has a positive correlation with the reflectance (Supplementary Fig. 6d). However, too high porosity may reduce the mechanical strength. Based on the trade-off of the optical and mechanical properties, ~ 60% porosity was adopted in the experiment design and radiative cooling application.

Figure 3d–f demonstrate the emissivity spectra of the PMMA$_{HPA}$ film in the infrared range of 2.5–20 μm at different polarization angles. $\bar{\varepsilon}_{LWIR}$ shows a regularly decreasing trend from 10° to 80°, mainly owing to the ordered periodic micropore array on the top surface of the PMMA$_{HPA}$ film. Excitingly, the average emissivity across the atmospheric window ($\bar{\varepsilon}_{LWIR}$) is greater than 0.95 over a wide polarization angle range from 10° to 80°, indicating a stable emitted heat flux through the atmospheric transparency window to the cold sink of outer space.

To investigate the influence of the hierarchical porous structure on the optical properties of the PMMA$_{HPA}$ film, we further compared it with three other types of PMMA films, nanopore PMMA$_{NP}$, monolayer micropore array PMMA$_{MPA}$ and pristine PMMA, as shown by the SEM images in Fig. 4a–d. Figure 4e shows that the PMMA$_{HPA}$ film presents the highest average solar reflectance ($\bar{\rho}_{HPA} = 0.95$) in UV-Vis-NIR wavelengths, while PMMA$_{NP}$ drops gradually to a lower average solar reflectance ($\bar{\rho}_{NP} = 0.74$) in the near-to-short wavelength infrared (NIR-to-SWIR) range (0.7–2.5 μm) because only these disordered nanopores are too small to effectively scatter such wavelengths. In contrast, PMMA$_{MPA}$ exhibits a low value at all wavelengths ($\bar{\rho}_{MPA} = 0.23$) because the scattering of uniform monolayer micropores is weak. Similar results can be observed for pristine PMMA ($\bar{\rho}_{prestine} = 0.10$) due to its high transparency. The emission spectra show that all four films have strong thermal emittances in the LWIR transparency window ($\bar{\varepsilon}_{HPA} = 0.98$, $\bar{\varepsilon}_{NP} = 0.96$, $\bar{\varepsilon}_{MPA} = 0.95$ and $\bar{\varepsilon}_{prestine} = 0.92$), as shown in Fig. 4f. However, the emissivity of PMMA$_{MPA}$ and pristine PMMA substantially drops in the range of mid-IR wavelengths compared to the PMMA$_{HPA}$ and PMMA$_{NP}$ films. This probably because that the existence of abundant hierarchically porous structure reduces the effective refractive index and leads to a more gradual

refractive index transition across the polymer-air interface than PMMA$_{MPA}$ and PMMA[17]. Thus, the impedance matching between the porous polymer and surrounding air is improved[48], which reduces the surface reflectance and results in a consistently higher emissivity for PMMA$_{HPA}$ or PMMA$_{NP}$ film in the mid-IR wavelengths. Besides, as we mentioned above, the porous structure with high specific surface area might increase the probability of infrared absorption through multiple diffuse reflection and enhances the emissivity.

The experimental results of four types of PMMA films were also theoretically verified by FDTD simulations (Supplementary Fig. 7)[49]. The numerical simulation results further reveal that hierarchical porous structure containing dual-scale nano/micro cavities would significantly improve the broadband scattering performance contrast to uniform nanoscale porous structure, especially in the range of NIR-to-SWIR, while the LWIR thermal emissivity of the two kinds of structures has negligible changes. The theoretical model of PMMA$_{MPA}$ verifies that the ordered micropores monolayer does not have enough high scattering coefficient, which matches well with the experimental result. We also simulated the reflectance spectra across the solar wavelengths of the PMMA$_{HPA}$ film with thickness of ~5 μm (effectively ~2 μm, Supplementary Fig. 8). Surprisingly, such thin film is sufficient to yield an efficient scattering ($\bar{\rho}_{HPA-2μm} = 0.35$), which further evidences the enhancement effect of hierarchical porous surface on the solar reflectance.

Besides, to verify the optical superiority of the periodic micropores array on the surface, we also investigated the optical properties of a hierarchically porous PMMA (PMMA$_{HP}$) film with loose-packed random micropores and nanopores (Supplementary Figs. 9a–c). The solar reflectance of the PMMA$_{HP}$ film drops 8% in the range of NIR-to-SWIR compared to PMMA$_{HPA}$ (Supplementary Fig. 9d), which contains about 5% of sunlight ($\bar{\rho}_{HP} = 0.90$). This suggests that the close-packed periodic arrangement of the hierarchical nano/microscale pores on the surface of our PMMA$_{HPA}$ can maximize both the surface area and the amount of scatters per unit and increase the overall scattering efficiency, especially in NIR-to-SWIR range, although this distribution of micropores on the surface does not influence the infrared emissivity (Supplementary Fig. 9e).

Furthermore, the PMMA$_{HPA}$ film was modified by fluorosilane to become superhydrophobic with a water contact angle (WCA) of ~156° (Supplementary Fig. 10) for stable durability in various atmospheric humidity conditions. Both the solar reflectance and infrared emissivity are at high levels and vary negligibly after fluorosilane treatment (Supplementary Figs. 11a and b). Even after accelerated weathering treatment for 480 h (each cycle including the UV irradiation at 310 nm wavelength with intensity of 0.71 W/m$^2$ at 60 °C for 4 h, followed by condensation at 50 °C for 4 h with UV lamps off), the PMMA$_{HPA}$ films exhibit no blistering, peeling, cracking and color changing. The porous morphology of our PMMA$_{HPA}$ films basically remains the same (Supplementary Figs. 12a and b). The ATR-FTIR spectra further demonstrate that the surface-modified PMMA$_{HPA}$ films before and after weathering treatment all have obvious absorption peaks of C=O at 1726 cm$^{-1}$, C–F at 1187 cm$^{-1}$, C–O–C at 1139 cm$^{-1}$ and C–Cl at 779, 746, 700, and 651 cm$^{-1}$ (Supplementary Fig. 12c), owing to the protection of fluorosilane molecules. Moreover, the constant WCAs in Supplementary Fig 12d after weathering treatment also indicate the excellent durability and potential applicability. The reflectance and emissivity spectra of the PMMA$_{HPA}$ films show slight fluctuations of solar reflectance ($\bar{\rho}_{solar} = 0.95 \pm 0.02$) and negligible variations in emissivity ($\bar{\varepsilon}_{LWIR} = 0.98 \pm 0.01$) with the accelerated weathering time (Supplementary Figs. 12e and f), which may be attributed to the inconspicuous yellowing effect in response to the UV irradiation. We also conducted the real outdoor exposure test for 40 days on a flat roof in Shanghai city and the results indicate the almost unchanged optical performance and WCAs (Supplementary Table S1).

**All-day continuous passive radiative cooling measurements of the PMMA$_{HPA}$ film**. The radiative cooling performances of the structural polymers (~160 μm effective thickness and 100 mm × 100 mm in size) were measured from 20:00 on 09 Oct. to 20:00 on 10 Oct. 2019 using a 24-h uninterrupted thermal measurement on a flat roof of a five-story building under a clear sky in Shanghai, China. As shown in Fig. 5a, the strong optical scattering of sunlight gives our PMMA$_{HPA}$ surface a matte and white appearance,

which can be further confirmed by the CIE chromaticity coordinate analysis (Fig. 5b). Under intense solar irradiance of ~930 W/m$^2$ and relative humidity of ~40% at noon (Fig. 5c), the real-time temperature tracking of the air and four types of PMMA films are shown in Fig. 5d. Evidently, although the pristine PMMA and PMMA$_{MPA}$ films can exhibit decreases in temperature by ~3.7 °C and ~4.9 °C at night, respectively, their temperatures dramatically rise to ~15.2 °C and ~14.1 °C above the ambient temperature at noon due to their transparency and translucency, respectively (Fig. 5e). The PMMA$_{NP}$ film can achieve a subambient cooling of ~6.5 °C during the night but maintains almost the same temperature as the ambient environment during midday. In contrast, the PMMA$_{HPA}$ film exhibits fabulous passive radiative cooling during both night and daytime. The average below-ambient temperature of the PMMA$_{HPA}$ film is ~8.2 °C during the night (between 6 p.m. and 6 a.m.) and ~6.0 °C during midday (between 11 a.m. and 2 p.m.). This cooling performance is on par with those in previous reports (Supplementary Table S2).

**Subambient PDRC performance in various geographical regions and climates**. Radiative cooling performance in real-world applications is substantially affected by the geographical regions and climates. For instance, the net radiative cooling power of the cooler is limited by the increased solar irradiation, humidity, cloud cover, local wind speed and ambient temperature[30,50]. Here, we further performed a series of experiments to evaluate and compare the effects of climates from different cities on the radiative cooling performance of our PMMA$_{HPA}$ films. Three cities, Xiamen city (Southern China, Coastal), Shanghai city (Eastern China, Coastal) and Xuzhou city (Northern China, Inland), were chosen as typical test locations due to their different topographic and meteorological characteristics (Supplementary Fig. 13). The climate characteristics in Xuzhou, Shanghai and Xiamen are temperate monsoon climate, subtropical monsoon climate and subtropical marine monsoon climate, respectively, which provide a remarkable difference in humidity for our measurements. As shown in Fig. 6a–i, under clear skies with comparable solar intensity and wind speed but various humidity conditions (~38% in Xuzhou, ~47% in Shanghai, and ~54% in

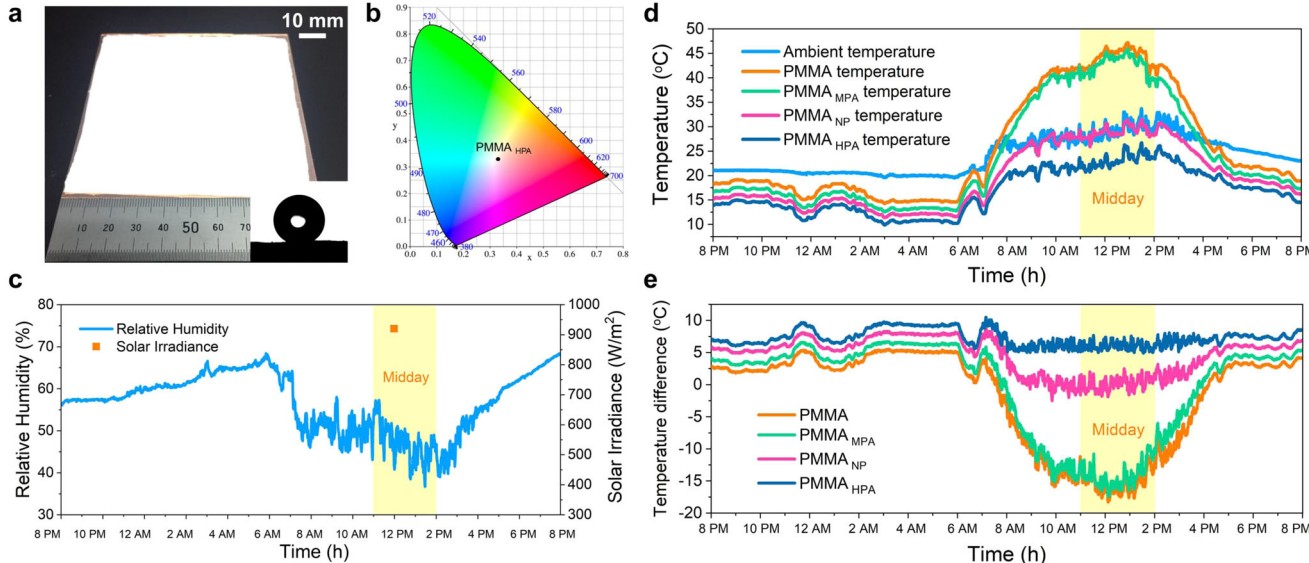

**Fig. 5 Twenty-four-hour continuous passive radiative cooling performance measurements. a** Photograph of the cooling PMMA$_{HPA}$ film showing its bright white appearance (inset, water contact angle image of PMMA$_{HPA}$). **b** CIE chromaticity coordinates of the cooling PMMA$_{HPA}$ film. **c** Relative humidity tracking of the air and solar irradiance at 12 PM on 10 Oct. 2019. **d** Temperature tracking of the air and PMMA films. **e** Temperature difference between the ambient and structural PMMA films.

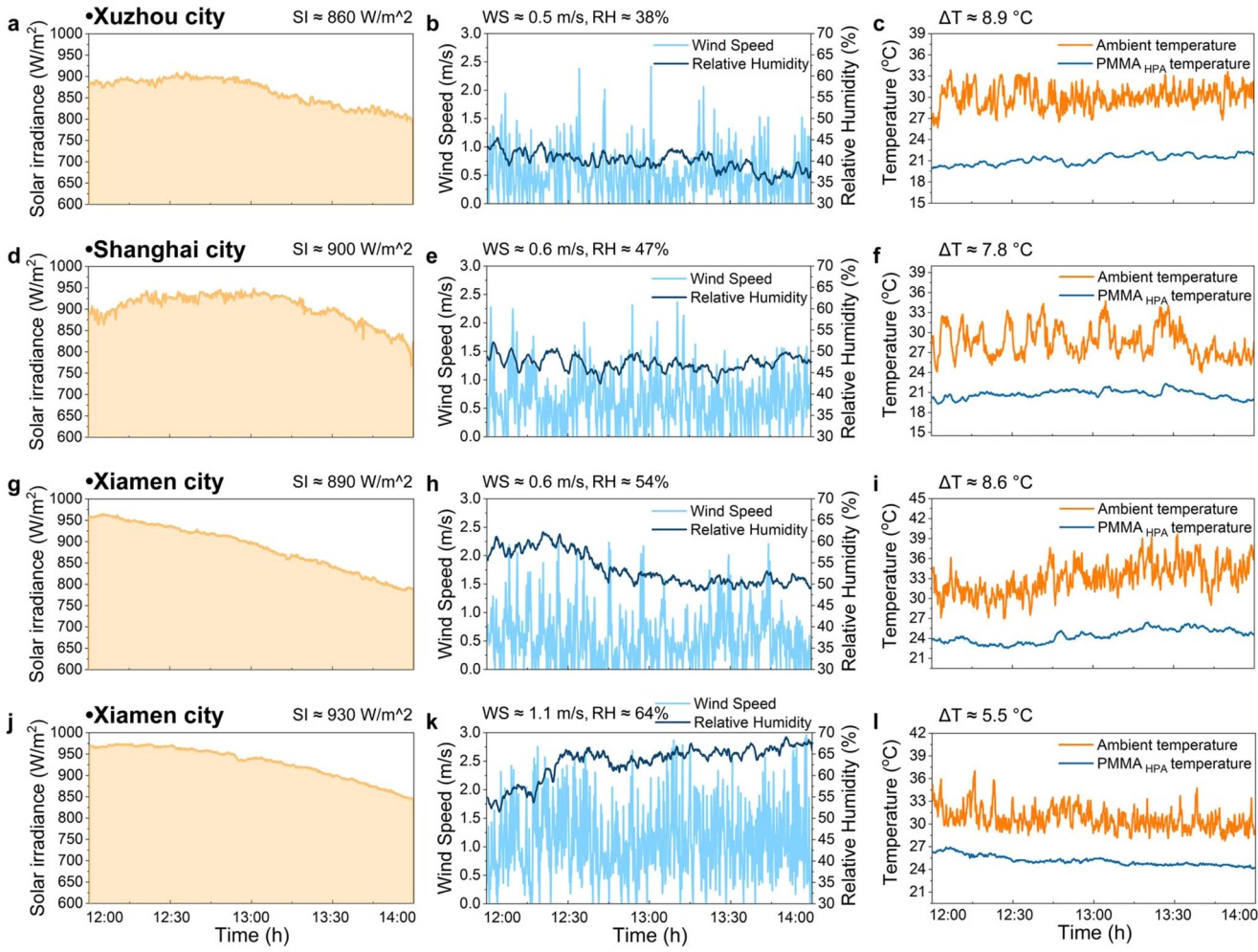

**Fig. 6 Passive daytime radiative cooling performance of the PMMA_HPA film in different locations and weathers. a–c** Solar irradiance, wind speed and relative humidity, and temperatures of the air and the PMMA_HPA film in Xuzhou city. **d–f** Solar irradiance, wind speed and relative humidity, and temperatures of the air and the PMMA_HPA film in Shanghai city. **g–i** Solar irradiance, wind speed and relative humidity, and temperatures of the air and the PMMA_HPA film in Xiamen city. **j–l** Solar irradiance, wind speed and relative humidity, and temperatures of the air and the PMMA_HPA film in Xiamen city.

Xiamen), the subambient cooling of ~8.9 °C, ~7.8 °C and ~8.6 °C at noontime is observed, separately. Even in the same location (e.g., in Xiamen city), increasing solar intensity (from ~890 W/m² to ~930 W/m²), wind speed (from ~0.6 m/s to ~1.1 m/s) and humidity (from ~54% to ~64%), the PMMA_HPA film still enables cooling up to ~5.5 °C below ambient temperature (Figs. 6g–l), which has not been reported previously[40,41]. Comparing Xuzhou and Xiamen city (Fig. 6a–c and j–l), we can conclude that increasing solar intensity (from ~860 W/m² to ~930 W/m²), wind speed (from ~0.5 m/s to ~1.1 m/s) and humidity (from ~38% to ~64%), the subambient cooling temperatures of PMMA_HPA film are indeed influenced (decreasing from ~8.9 °C to ~5.5 °C). The primary driver of the good performance in various geographical regions and climates should be the synergistic result of visible white (high solar reflectance) and infrared black (high infrared emissivity in both the 1st and 2nd atmospheric transparency window) that greatly minimizes the absorbing solar irradiance and the thermal radiation emitted by the atmosphere. In addition, we investigated the effect of surface modification on the daytime radiative cooling performance of the PMMA_HPA film. The results show that the PMMA_HPA films with and without surface modification can achieve subambient cooling temperatures of ~6.9 °C and ~6.5 °C, respectively (Supplementary Fig. 14). While the PMMA_HPA film without surface modification also demonstrates

excellent passive radiative cooling behavior, the fluorosilane treatment enables stable performance by restricting the effect of moisture and water under different levels of humidity.

**Passive radiative cooling power measurements of the PMMA_HPA film.** When used in a building roof or external siding, our PMMA_HPA films can achieve all-day passive radiative cooling through reflecting sunlight and radiating heat to the cold outer space under a clear sky (Fig. 7a). Figure 7b schematically shows the direct thermal measurement system based on the net cooling equation. To further demonstrate the PDRC capability of the cooling PMMA_HPA, we adopted a feedback-controlled heating system to measure its radiative cooling power during the midday (Fig. 7c). This feedback-controlled heating system maintains the surface temperature of the PMMA_HPA film at the measured ambient temperature to minimize the impact of conductive and convective heat losses (Fig. 7d). Promisingly, the PMMA_HPA film attains an average cooling power of ~85 W/m² under solar intensity of ~900 W/m² in April in Shanghai (Fig. 7e).

Figure 8a, b present the net cooling power during the nighttime and daytime calculated using the radiative cooling theoretical model, respectively. More details of this model are given in the "Methods" section. The power of solar radiation is set to

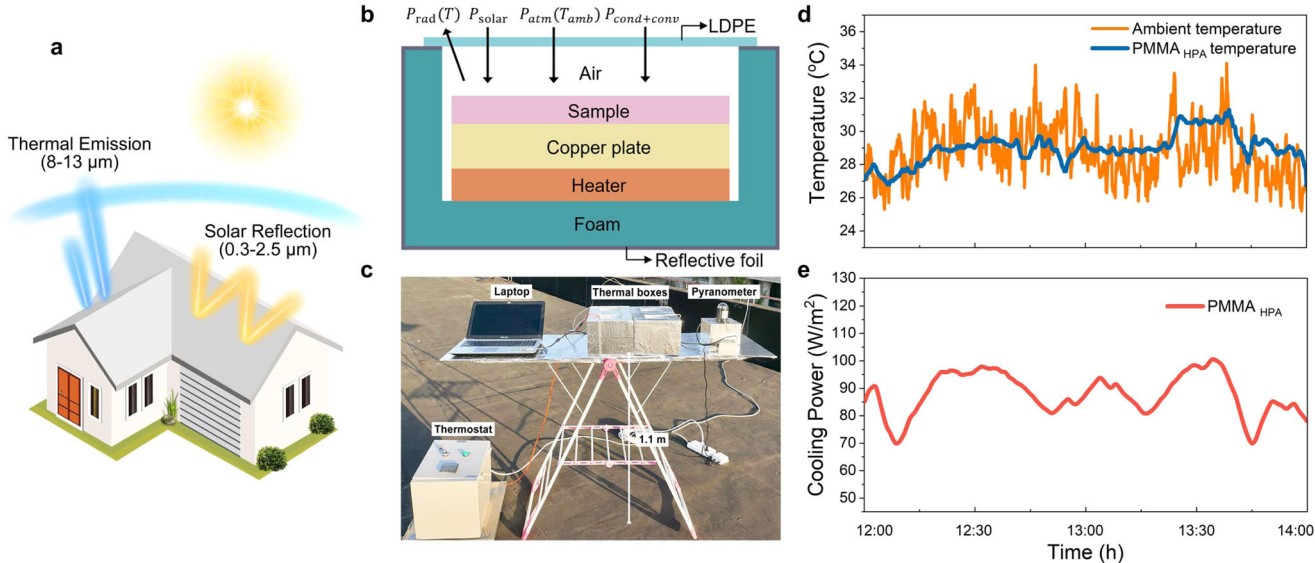

**Fig. 7 Net cooling power of the PMMA$_{HPA}$ film during the midday. a** Schematic of the basic principles of PDRC. When used in a building roof or external siding, the PMMA$_{HPA}$ film exhibits high solar reflectance and high infrared emissivity. **b** Two-dimensional schematic drawing of the thermal box apparatus with a feedback-controlled heater. The heater maintains the sample surface temperature at that of the ambient environment, minimizing convective and conductive heat losses. **c** Photo of the experimental apparatus on a rooftop in Shanghai, China. **d** Temperature tracking of the ambient and the PMMA$_{HPA}$ film under solar intensity of ~ 900 W/m² and relative humidity of ~ 44% on April 23, 2020. **e** Continuous measurement of the radiative cooling power of the PMMA$_{HPA}$ film during the midday.

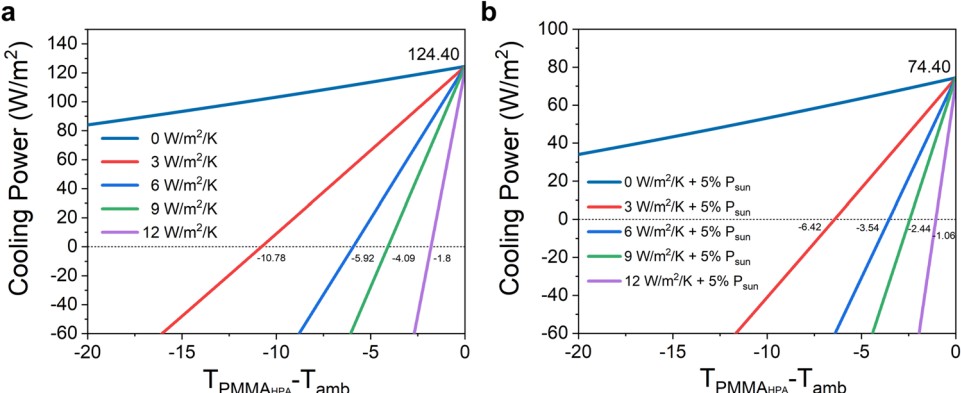

**Fig. 8 Calculated net cooling power with software MATLAB based on the theoretical simulation. a** Calculated net cooling power during the nighttime. **b** Calculated net cooling power during the daytime. The variable $h_c$ is a combined nonradiative heat coefficient. Values of 0, 3, 6, 9, and 12 for $h_c$ are used in the calculations. For daytime calculations, 5% solar power absorption is considered.

approximately 1000 W/m² for simplicity, and the ambient temperature $T_{amb}$ is assumed to be 298.15 K in both cases. A maximum cooling power of 124.40 W/m² can be achieved for nighttime operation. For daytime operation, the calculated maximum net cooling power is 74.40 W/m² at thermal equilibrium, which is lower than the measured daytime cooling power due to the fluctuations in the ambient conditions, the uncertainty in the measurements and the theoretical model approximations.

In summary, we have demonstrated and fabricated a hierarchically structured PMMA film with a dense micropore monolayer array and randomly distributed nanopores for highly efficient all-day passive subambient radiative cooling in various geographical locations and climates. Our structural polymer film exhibits sufficiently high solar reflectance and thermal emittance owing to its abundant periodic scattering micropores embedded with random nanopores and ideal intrinsic properties. Without needing any silver or aluminum reflectors, our cooling structural

PMMA film realizes an average below-ambient temperature ~8.2 °C during the night and ~6.0 °C to ~8.9 °C during midday, and promisingly ~5.5 °C even under solar intensity of ~930 W/m² and relative humidity of ~64% in hot and moist subtropical marine monsoon climate, which is really an all-day and all-climate PDRC system. And the superhydrophobized PMMA$_{HPA}$ film can ensure cooling performance durability by eliminating the effect of moisture and water under different levels of humidity. This study has revealed the effects of micropores, nanopores and their arrangement on optical performance, which may provide deep insight into the crucial roles of various pores and their array in solar reflectance and thermal emittance and help us design and fabricate more efficient all-day passive subambient radiative cooling materials and systems.

## Methods

**Fabrication of PMMA$_{HPA}$.** Monodisperse 5 μm SiO$_2$ microspheres were placed on top of a polydimethylsiloxane (PDMS)-coated glass sheet and rubbed with another

PDMS substrate with slight palm pressure along a randomly chosen direction according to our reported method[43,44]. After being rubbed for 5 s, the SiO$_2$ microspheres had assembled into hexagonally close-packed monolayers on the PDMS surfaces. A 200 nm SiO$_2$ nanosphere dispersion in acetone was added to PMMA to make a dispersion of SiO$_2$-acetone-PMMA (1:10:1 mass ratio) under magnetic stirring at 50 °C for 2 h. This dispersion was then drop-cast onto the monolayer SiO$_2$ template. After the solvent was fully evaporated, a freestanding PMMA/SiO$_2$ composite film was obtained by peeling the coating off the smooth surface. After removal of the SiO$_2$ template and nanospheres with 2-5 vol% hydrofluoric acid aqueous solution, PMMA$_{HPA}$ films with various effective thicknesses of $2 \pm 0.5\ \mu m$ to $160 \pm 5\ \mu m$ were obtained by casting different amounts of solution. For comparison, PMMA$_{NP}$ film was fabricated by the same procedure above without using the monolayer SiO$_2$ template, PMMA$_{MPA}$ film was obtained without SiO$_2$ nanospheres, and the pristine non-porous PMMA film was prepared at an acetone-PMMA (10:1 mass ratio) without using SiO$_2$ particles. In addition, PMMA$_{HP}$ film was fabricated using random loose-packed monolayer 5 $\mu m$ SiO$_2$ templates and 200 nm SiO$_2$ nanosphere. All the films above have the same effective thickness by controlling the PMMA mass and film-forming substrate size.

**PMMA$_{HPA}$ surface modification.** To make superhydrophobic PMMA$_{HPA}$ film, the sample was treated with 1H,1H,2H,2H-perfluorooctyltrichlorosilane (PFOTS) via a chemical vapor deposition (CVD) method. In detail, the PMMA$_{HPA}$ film and a vial containing 1% PFOTS/ethanol solution were placed in a sealed vacuum desiccator. The desiccator was immediately pumped to vacuum for 15 min and placed at room temperature for 24 h for fluorination. The modified PMMA$_{HPA}$ film was then baked in an oven at 80 °C for 1 h to remove the excessive PFOTS.

**Characterization**

*Optical characterization of the cooling structural polymers.* The spectral reflectance ($\rho$ ($\lambda$)) and transmittance ($\tau$ ($\lambda$)) in the ultraviolet, visible and near-infrared (0.3–2.5 $\mu m$) wavelength ranges were separately determined in an UV-Vis-NIR spectrophotometer (Hitachi, U-4100, Japan) with a polytetrafluoroethylene integrating sphere. The $\rho$ ($\lambda$) and $\tau$ ($\lambda$) in the mid-infrared wavelength ranges were characterized in an FTIR spectrometer (Nicolet 6700, Thermo Fisher Scientific, USA) equipped with a gold integrating sphere. We put one black substrate behind the samples during spectral reflectance measurement to eliminate the reflectance contribution of the substrate. The average value of more than five parallel measurements on different sites of the film was reported and the error bars represent the standard deviation. A polarizer was used in the FTIR spectrometer equipped with a smart diffuse reflectance accessory to measure the reflectance over a range of polarization angles ($\theta$) from 10° to 80°. For any object at thermal equilibrium, the spectral absorptivity ($\alpha$ ($\lambda$)) and emissivity ($\varepsilon$ ($\lambda$)) must be equal according to Kirchhoff's law; thus, $\varepsilon$ ($\lambda$) was calculated as $\varepsilon$ ($\lambda$) = 1−$\rho$ ($\lambda$)−$\tau$ ($\lambda$)[51,52]. The angular reflectance spectra in the wavelength range of 0.4–1.1 $\mu m$ at different incidence angles were measured by an angle-resolved photonic spectral system (R1, Ideaoptics Technology Ltd., China). Refractive index and extinction coefficient measurements in the wavelength range of 0.2–14 $\mu m$ at 60° were taken for pristine PMMA films using a V-VASE and an IR-VASE ellipsometer (J. A. Woollam, USA).

*Theoretical model of the radiative cooling performance.* When the structural polymer films are exposed to a clear sky, they are influenced by the solar irradiance and atmospheric downward thermal radiation. Meanwhile, heat can be transferred from the ambient surroundings to the polymers via conduction and convection due to the temperature difference between the cooling polymers and the ambient environment. To achieve PDRC, the device must satisfy very stringent constraints as dictated by the power balance equation. The net cooling power $P_{cool}$ of the structural polymers is expressed as[26]:

$$P_{cool}(T) = P_{rad}(T) - P_{atm}(T_{amb}) - P_{solar} - P_{cond+conv} \tag{1}$$

where $T$ is the surface temperature of the structural polymers and $T_{amb}$ is the ambient temperature. $P_{rad}(T)$ is the power radiated by the structural polymers, and $P_{atm}(T_{amb})$ is the absorbed atmospheric thermal radiation at $T_{amb}$. $P_{solar}$ is the incident solar irradiation absorbed by the structural polymers, and $P_{cond+conv}$ is the power lost due to convection and conduction. The net cooling power defined in Eq. (1) can reach a high value by increasing the radiative power of the structural polymers and reducing either the solar absorption or parasitic heat gain[34]. These parameters can be calculated by the following equations[26,53–55]:

$$P_{rad}(T) = A \int d\Omega \cos\theta \int_0^\infty d\lambda I_{BB}(T, \lambda)\varepsilon(\lambda, \theta) \tag{2}$$

$$P_{atm}(T_{amb}) = A \int d\Omega \cos\theta \int_0^\infty d\lambda I_{BB}(T_{amb}, \lambda)\varepsilon(\lambda, \theta)\varepsilon_{atm}(\lambda, \theta) \tag{3}$$

$$P_{solar} = A \int_0^\infty d\lambda \varepsilon(\lambda, \theta_{solar})I_{AM1.5}(\lambda) \tag{4}$$

$$P_{cond+conv}(T, T_{amb}) = Ah_c(T_{amb} - T) \tag{5}$$

where $A$ is the surface area of the radiative cooler. $\int d\Omega = 2\pi \int_0^{\pi/2} d\theta \sin\theta$ is the angular integral over a hemisphere. $I_{BB}(T, \lambda) = \frac{2hc^2}{\lambda^5} \frac{1}{e^{hc/(\lambda\kappa_B T)} - 1}$ is the spectral radiance of a blackbody at temperature $T$. $h$ is Planck's constant, $\kappa_B$ is the Boltzmann constant, and $c$ is the speed of light. $\varepsilon(\lambda, \theta)$ is the directional emissivity of the surface at wavelength $\lambda$. $\varepsilon_{atm}(\lambda, \theta) = 1 - \tau(\lambda)^{1/\cos\theta}$ is the angle-dependent emissivity of the atmosphere; $\tau(\lambda)$ is the atmospheric transmittance in the zenith direction. $P_{rad}(T)$ and $P_{atm}(T_{amb})$ are determined by both the spectral data of the structural polymers and the emissivity spectrum of the atmosphere according to MODTRAN of Mid-Latitude Summer Atmosphere Model[36,56]. In Eq. (4), the solar illumination is represented by the AM1.5 spectrum ($I_{AM1.5}(\lambda)$). For our PMMA$_{HPA}$ film, approximately 95% of the input solar power can be reflected, and thus, the 5% absorption of the solar irradiance will reduce the net cooling power. In Eq. (5), $h_c = h_{cond} + h_{conv}$ is a combined nonradiative heat coefficient that captures the collective effect of conductive and convective heating, which can be limited to a range between 0 and 12 W/m$^2$/K.

The average solar reflectance ($\bar{\rho}_{solar}$) is defined as[17]:

$$\bar{\rho}_{solar} = \frac{\int_{0.3\mu m}^{2.5\mu m} I_{solar}(\lambda) \cdot \rho_{solar}(\lambda, \theta)d\lambda}{\int_{0.3\mu m}^{2.5\mu m} I_{solar}(\lambda)d\lambda} \tag{6}$$

where $\lambda$ is the wavelength of incident light in the range of 0.3–2.5 $\mu m$, $I_{solar}(\lambda)$ is the normalized ASTM G173 Global solar intensity spectrum, and $\rho_{solar}(\lambda, \theta)$ is the surface's angular spectral reflectance. The average emittance ($\bar{\varepsilon}_{LWIR}$) in the LWIR atmospheric transmittance window is defined as:

$$\bar{\varepsilon}_{LWIR} = \frac{\int_{8\mu m}^{13\mu m} I_{BB}(\lambda) \cdot \varepsilon_{LWIR}(\lambda, \theta)d\lambda}{\int_{8\mu m}^{13\mu m} I_{BB}(\lambda)d\lambda} \tag{7}$$

where $I_{BB}(\lambda)$ is the spectral intensity emitted by a blackbody and $\varepsilon_{LWIR}(\lambda, \theta)$ is the surface's angular spectral thermal emittance in the range of 8–13 $\mu m$.

*Thermal measurements of cooling temperature and cooling power with a feedback-controlled heater.* We designed a feedback-controlled program to minimize both conductive and convective heat exchange to the structural polymers under strong solar irradiance based on Eq. (1)[29]. Our thermal box consisted of insulation foam covered by a layer of reflective foil. A 10-$\mu m$-thick transparent low-density polyethylene film was used to seal the thermal box and served as a wind shield. Moreover, a PMMA$_{HPA}$ film with a size of 100 mm × 100 mm × 160 $\mu m$ was placed on a 100 mm × 100 mm × 1 mm thick copper plate attached to a 100 mm × 100 mm × 0.2 mm thick Kapton heater. The Kapton heater was feedback-controlled by an ambient temperature-responsive thermostat to maintain the PMMA$_{HPA}$ film at ambient temperature and accurately assess the cooling power. The apparatus was elevated 1.1 meters above the ground to avoid heat conduction from the ground to the thermal box. An adhesive resistant temperature detector was directly mounted on the back surface of the polymer film to detect real-time temperature of the sample, which was continuously recorded by a datalogging thermometer with an uncertainty of $\pm$ 0.1 °C (CENTER309, CENTER corp, Taiwan, China). For comparison, a temperature detector was mounted outside the box to detect real-time temperature of the ambient. A relative humidity (RH) data logger with an accuracy of $\pm$ 0.1% RH (GSP-8, Elitech corp, China) was placed near the samples to measure the relative air humidity. The solar irradiation outside the box was simultaneously recorded using a datalogging solar radiometer with an accuracy of $\pm$ 5% (TES1333R, TES Electrical Electronic corp. Taiwan, China). It is worth mentioning that the sunlight our PMMA films received is ~10% less than the pyranometer measured due to the polyethylene cover on the thermal box. The wind speed around our thermal boxes was measured using a digital anemometer with an accuracy of $\pm$2.5% (AS856, Smart Sensor corp, China). All weather data were automatically tracked every 10 s. The heater was switched on to test the radiative cooling power, but the heater and copper plate were removed to demonstrate the subambient cooling performance. As a control, we exposed the PMMA$_{NP}$, PMMA$_{MPA}$ and pristine PMMA films to the sky while mounted in the same apparatus to compare the subambient cooling performances. Demonstrations of the cooling performance of different types of PMMA films during both day and night were carried out under a clear sky with a relative humidity of ~ 40% at noon in on a flat roof of a five-story building at Fudan University, Shanghai, China on October 09, 2019. The daytime cooling performance of the PMMA$_{HPA}$ film in different locations and climates was conducted in April and May, 2020, such as Xiamen city (Southern China, Coastal, 24° 26' 57" N, 118° 3' 35" E), Shanghai city (Eastern China, Coastal, 31° 18' 22" N, 121° 30' 17" E) and Xuzhou city (Northern China, Inland, 34° 27' 55" N, 117° 0' 51" E).

## Data availability

All data needed to evaluate the conclusions in the paper are presented in the paper and/or the Supplementary information. Additional data related to this paper may be requested from the authors.

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

## Acknowledgements

We appreciate the financial support provided for this research by the National Key Research and Development Program of China (2017YFA0204600 and 2020YFE0100300) and the National Natural Science Foundation of China (52033003 and 51721002).

## Author contributions

L.W., T.W., and M.C. conceived the concept and designed the research. T.W. and Y.W. conducted the experiments. T.W. and L.S. conducted the FDTD simulations. T.W. created the schematics. L.W. and T.W. wrote the manuscript. All authors including X.H. discussed the results and commented on the manuscript.

## Competing interests

The authors declare no competing interests.
