## [Peer Review File · Nature Communications]

Editorial Note: The figure on page 19 in this Peer Review File has been amended to remove third-party material where no permission to publish could be obtained.

REVIEWER COMMENTS

Reviewer #1 (Remarks to the Author):

This paper reports a porous PMMA film with impressive radiative cooling properties. The solar reflectance is 95% and the thermal emittance reaches 0.98. The performance is among the best values reported. I have several questions regarding details inside. I can consider acceptance if all questions below are well addressed.

1. It is surprising that a single layer of 5 μm pores can enhance solar reflectance in such a dramatic way, since the scattering of a single layer is weak, even for wavelengths comparable to the pore size. I strongly suggest authors run optical simulation to justify the reflectance, including both high reflectance, and also how the 5 μm particle enhances reflectance. My personal experience is that it is difficult to achieve such high reflectance by the design shown here.
2. Can authors comment on the complication of the process, since HF etching does not seem to be an economically viable approach? How is the cost?
3. The UV stability of PMMA should be discussed more. This is a major concern for all acrylic-based paint.

Reviewer #2 (Remarks to the Author):

This work presents a new hierarchically porous PMMA selective emitter for all-day passive radiative cooling. By combining the favorable intrinsic optical properties of PMMA (low solar and high infrared absorption) with the scattering of a hierarchical porous structure, the proposed selective emitter achieved high solar reflectance and infrared emittance, enabling daytime sub ambient cooling up to 8.9 $^{\circ}\text{C}$ below the ambient and ambient temperature cooling power up to 85 W/m^2 . While the selective emitter performance is impressive and the results are well presented, the work's novelty as well as robust theoretical modeling and understanding of the mechanisms behind the film optical properties are limited. As such, we believe the paper is better suited for publication in another journal than Nature Communications. Below are comments and suggestions which we hope will help further improve the quality of the paper.

1. Abstract: The cost, scalability and impact of the work seem to be missing. Many papers on selective emitters for all day passive radiative cooling have already been published in the literature. Clearly mentioning the differences and contributions of this work over past work can help better understand the impact and novelty of this work.
2. In the introduction, the authors mention that passive radiative cooling could be used for geoengineering. While radiative cooling can provide passive cooling for building, its use for geoengineering might not make sense given the massive scale required.
3. It is also not clear or established that radiative cooling can completely replace compression-based cooling systems. Please correct accordingly.
4. The authors highlight the high manufacturing cost and large-scale production limits of past works, but they do not address these topics with their own proposed emitter. How is the proposed film better than previous work regarding these aspects? What is the estimated cost? Several papers have already proposed low-cost and scalable emitters for radiative cooling, as well as companies which are already using such products. The authors should more carefully look at the existing literature's work and appropriately acknowledge them.
5. In the introduction, the authors also suggest that their film works well in arid and humid climates as opposed to other emitters. What fundamental difference does the proposed film have with other emitters that allows it to perform better in humid conditions? One could argue that the proposed film would perform worse in humid weather conditions compared to some other selective emitters that only emit in the 8-13 μm windows and have high reflectivity outside that range. In higher humidity environment, the atmosphere is only transparent between 8-13 μm and thus any absorption outside that wavelength range decreases the cooling performance. The film proposed here has a relatively constant emissivity in the infrared, as opposed to an optimized 8-13 μm selective emissivity. Overall, we do not think the proposed film should have better performance in humid weather than previously demonstrated emitters based on its optical properties.

6. How was the micro and nano pores size chosen? The current manuscript is missing information on the choice of these parameters and their optimization. The manuscript would strongly benefit from a theoretical model that estimates the scattering and absorption coefficients of the film based on the pore sizes and density. Such a model would also help highlight the benefits of using a hierarchical porous structure over a uniform size porous structure by showing an improvement (if it is the case) in scattering albedo in the solar spectrum when using a hierarchical porous network. The model would also give a better understanding of what is the role of the porous structure in the infrared and insights on how to improve the film further.
7. In SI Fig. 3, why are the curves for 12 μm to 100 μm thicknesses in the 4-6 μm range all in the same range of transmittance?
8. The calculation of the porosity in the SI is confusing. How do you get the thickness of the non-porous PMMA sample? Do you compress that sample until there is no more porosity or do you make another PMMA film with no porosity? Why wasn't a density-based porosity measurement used?
9. Fig. 3b and c: Are the values of reflectance/emittance interpolated between each experimental measurement?
10. Fig. 3c, what happened at a thickness of 40 μm ? Is that an anomaly or an expected behavior?
11. Fig. 3d-f. The polarization angle is changed. Do the authors actually refer to the incident angle of the light on the sample or the actual polarization of light? The two are different and should not be confused here.
12. The authors mention that the atmospheric window emissivity is greater than 0.95 over a wide polarization angle range. Assuming they the polarization angle range refers to the incident angle, is it really good to have high emissivity at large incident angles considering that the atmospheric transmittance decreases with increasing zenith angle?
13. The comparison between the PMMA_hpa film and the three other PMMA films is interesting, but it is not clear if the comparison is fair. Is the thickness of each film kept constant? Does the ordered micropores monolayer not achieve high solar reflectivity only because it is not thick enough or because the micropores do not have a high enough scattering coefficient? Again, a theoretical model might help here to compare the different films or pore structures/sizes for similar thicknesses and conditions.
14. What do the authors mean by "contact areas with light" at line 221?
15. An accelerated weathering test was performed for 480h. What conditions were these made in? To what equivalent real outdoor time would that correspond to?
16. Was there a substrate to deposit the film on during the stagnation temperature measurements? What did the setup look like during this test?
17. "This cooling performance is on par with or exceeds those in previous reports". It would be more accurate to say that it is on par with those in previous reports as other works has demonstrated higher performance.
18. Add reference for line 287-289.
19. Line 298: The film is tested in different cities with different RH. The values however only change by 16% absolute RH. How significant is that?
20. Fig. 8: What atmospheric model was used for the cooling power curves? The high nighttime cooling power of nearly 140 W/m^2 suggests that a very low humidity environment was used for the modeling, which might differ with the experimental locations.

Reviewer #3 (Remarks to the Author):

Passive radiative cooling with no energy input is an appealing technology to meet the demand of cooling. Therefore, recently people are developing all kinds of materials or structures to enhance the cooling performance. The authors proposed a polymer (i.e., PMMA) with a hierarchically porous array. It can achieve high solar reflectance and thermal emittance, and therefore, could be employed to realize passive daytime radiative cooling. This manuscript should be strengthened with more clarification, experimental results and further analysis on the following details.

1. The structural polymer proposed by the authors is also one kind of porous polymer mentioned in the introduction of the manuscript. What are the advantages of the proposed structural polymer compared with the ones reported previously?
2. The authors mentioned the surface modification by fluorosilane, but did not show the detailed

experiment process. It is also important to compare the reflectance, emissivity and the radiative cooling performance before and after the surface modification.

3. The details of the weathering test are not shown, including how heat, water, oxygen and UV radiation were applied on the polymer.

Point-by-Point to Reviewers' Comments

To Reviewer #1:

General comments: This paper reports a porous PMMA film with impressive radiative cooling properties. The solar reflectance is 95% and the thermal emittance reaches 0.98. The performance is among the best values reported. I have several questions regarding details inside. I can consider acceptance if all questions below are well addressed.

General answers: Thanks the Reviewer very much for his or her positive comments and highly valuable suggestions. We are trying our best to revise our manuscript according to these comments and suggestions.

Q1: It is surprising that a single layer of 5 μm pores can enhance solar reflectance in such a dramatic way, since the scattering of a single layer is weak, even for wavelengths comparable to the pore size. I strongly suggest authors run optical simulation to justify the reflectance, including both high reflectance, and also how the 5 μm particle enhances reflectance. My personal experience is that it is difficult to achieve such high reflectance by the design shown here.

A1: According to the Reviewer's suggestion, we have conducted Finite-Difference Time-Domain (FDTD) simulations to explain the optical mechanisms behind the structural PMMA films. Due to computational constraints and overlong computing time, we used two dimensional (2D) models for numerical calculations and reduced the effective thickness of the structural PMMA films. Please see the blue words in Supplementary pages 4-5.

We designed four types of PMMA models and kept their effective thickness consistent ($\sim 80 \mu\text{m}$). The simulated reflectance results of the four types of PMMA films are in good agreement with the experimental data in tendency (see Fig. 4e and the added Supplementary Fig. 7a), further revealing that the hierarchical porous structure containing dual-scale nano/micro cavities on the surface can significantly improve the broadband scattering performance compared to uniform nanoscale porous structure, especially in the range of NIR-to-SWIR. In addition, the PMMA_{HPA} film with only $\sim 80 \mu\text{m}$ thickness presents a high average solar reflectance ($\bar{\rho}_{solar} = 0.88$), which agrees with the measured reflectance result (see Fig. 3a and the added Supplementary Fig. 7a).

As the Reviewer said, the scattering of monolayer micropores is weak, which can be indeed seen in our experimental and theoretical results (see PMMA_{MPA} curves in Fig. 4e and the added Supplementary Fig. 7a). However, after the embedded with abundant randomized nanopores, the scattering efficiency considerably increases (see PMMA_{HPA} curves in Fig. 4e and the added Supplementary Fig. 7a). We also simulated the reflectance spectra across the solar wavelengths of the PMMA_{HPA} film with thickness of $\sim 5 \mu\text{m}$ (effectively $\sim 2 \mu\text{m}$, the added Supplementary Fig. 8). Surprisingly, such thin film is also sufficient to yield an efficient scattering ($\bar{\rho}_{HPA-2 \mu\text{m}} = 0.35$). Thus we can conclude that the hierarchical porous structure with a micropore array on the

film surface combined with abundant randomized nanopores plays crucial roles in enhancing the solar reflectance, which can be indirectly evidenced by the recently published paper (Zhang, H. *et al. P. Natl. Acad. Sci. USA.* **117**:202001802 (2020)).

More results and discussions have been added. Please see the blue words in page 13.

Q2: Can authors comment on the complication of the process, since HF etching does not seem to an economically viable approach? How is the cost?

A2: Thanks for the kind suggestion. Although HF etching seems not to be a very economical method, here the SiO₂ nanospheres and microspheres were removed by etching in 2-5 vol% hydrofluoric acid aqueous solution for only 1 h, and each HF aqueous solution can be recycled for more than ten times. In this study, we would just report a hierarchically structured polymer prototype with a micropore array on the surface combined with random nanopores for highly efficient day- and nighttime passive subambient radiative cooling. People may use other pore-creating methods to fabricate such structured polymer films as highly efficient PDRC for practical applications. PMMA is about 3.5 \$/kg, considerably cheaper than the previously reported polymers, e.g., PVDF and TPX (~ 29.6 \$/kg and ~ 5.5 \$/kg, respectively).

Q3: The UV stability of PMMA should be discussed more. This is a major concern for all acrylic-based paint.

A3: According to the Reviewer's suggestion, we have conducted the accelerated weathering test for 480 h and outdoor exposure test for 40 days to demonstrate the durability and UV stability of our PMMA_{HPA} film. After 480 h weathering treatment (each cycle including the UV irradiation at 310 nm wavelength with intensity of 0.71 W/m² at 60 °C for 4 h, followed by condensation at 50 °C for 4 h with UV lamps off), the PMMA_{HPA} films exhibit no blistering, peeling, cracking and color changing. The good durability and UV stability is due to the surface modification by fluorosilane. After outdoor exposure test for 40 days, our PMMA_{HPA} film still shows the same optical performance and WCAs as its original value. More discussions have been added. Please see the blue words in page 15 and the added Supplementary Table S1.

To Reviewer #2:

General comments: This work presents a new hierarchically porous PMMA selective emitter for all-day passive radiative cooling. By combining the favorable intrinsic optical properties of PMMA (low solar and high infrared absorption) with the scattering of a hierarchical porous structures, the proposed selective emitter achieved high solar reflectance and infrared emittance, enabling daytime sub ambient cooling up to 8.9 °C below the ambient and ambient temperature cooling power up to 85 W/m². While the selective emitter performance is impressive and the results are well presented, the work's novelty as well as robust theoretical modeling and understanding of the mechanisms behind the film optical properties are limited. As such, we believe the paper is better suited for

publication in another journal than Nature Communications. Below are comments and suggestions which we hope will help further improve the quality of the paper.

General answers: We appreciate the Reviewer very much for his or her encouraging comments. Considering the Reviewer's suggestion, we have conducted Finite-Difference Time-Domain (FDTD) simulations to investigate the influence of pore size, porosity and structures on the optical performance and explained the optical mechanisms behind the films. Moreover, we have compared the PDRC performances of our device with other state-of-the-art emitters to illustrate the main features and innovations of this work.

Q1: Abstract: The cost, scalability and impact of the work seem to be missing. Many papers on selective emitters for all day passive radiative cooling have already been published in the literature. Clearly mentioning the differences and contributions of this work over past work can help better understand the impact and novelty of this work.

A1: Thanks the Reviewer very much for his or her constructive comments and suggestions. First, with regard to the scalability, this work and our previous reported work both adopted the unidirectional rubbing assembly method and templating method, which are feasible for large-scale manufacture without much finesse. (Fan, W. *et al. Sci. Adv.* **5**, eaaw8755 (2019); Wu, Y. *et al. ACS nano* **12**, 10338-10346 (2018).). Here, we prepared several PMMA_{HPA} films with the size of 100 mm × 100 mm for the radiative cooling measurements (Fig. 5a).

Second, with regard to the cost, we selected one of the most prevalent and low-cost polymers, (PMMA, ~ 3.5 \$/kg). Although the fabrication process of the PMMA_{HPA} film is not very economical (by etching in 2-5 vol% hydrofluoric acid aqueous solution), this process takes only 1 h, and each HF aqueous solution can be recycled for more than ten times. Moreover, in this study, we would just report a hierarchically structured polymer prototype with a micropore array on the surface combined with random nanopores for highly efficient day- and nighttime passive subambient radiative cooling. People may use other pore-creating methods to fabricate such structured polymer films as highly efficient PDRC for practical applications. More discussions have been added. Please see the blue words in page 8.

Third, we have compared the PDRC performances of our device with other state-of-the-art radiative coolers, please see the added Supplementary Table S2. The main features and innovations of our work are as follows:

(1) Our PMMA_{HPA} film has a hierarchically structure with a close-packed micropore array on the surface combined with abundant random nanopores by a templating method, which is reported for the first time. Importantly, these micropores and nanopores and their arrangements can be precisely controlled by templating colloidal spheres. Accordingly, this study and the hierarchically porous structured PMMA model may provide deep insight into the crucial roles of micropores and nanopores in solar reflectance and thermal emittance and help us design and fabricate more

efficient all-day passive subambient radiative cooling materials and systems. Please see the blue words in pages 4 and 5.

(2) Because of the unique structure and favorable intrinsic optical properties, the as-obtained PMMA_{HPA} film demonstrates excellent $\bar{\rho}_{solar}$ and $\bar{\epsilon}_{LWIR}$ and superb subambient radiative cooling performance in various geographical regions and climates.

(3) As one of the most widely used and low-cost polymers, PMMA is very cheap (~3.5 \$/kg) and selected as our cooling radiator. Pristine PMMA film is highly transparent with negligible extinction coefficient in the solar wavelengths as well as multiple extinction peaks within the LWIR window, which is responsible for outstanding PDRC performance. The majority of current polymer emitters, such as PVDF and TPX, are also conducive to radiative cooling, but they seem to be more expensive (~29.6 \$/kg and ~5.5 \$/kg, respectively).

Some discussion has been added, please see the blue words in pages 4 and 5. We hope our study can help people to further deep understand the roles of micropores and nanopores and their arrangement in optical properties and to find better PDRC materials and systems for practical uses.

Q2: In the introduction, the authors mention that passive radiative cooling could be used for geoeengineering. While radiative cooling can provide passive cooling for building, its use for geoeengineering might not make sense given the massive scale required.

A2: Thanks for the kind suggestion. This “One alternative geoeengineering approach” has been revised to “One possibly alternative approach”. Please see the blue words in page 3.

Q3: It is also not clear or established that radiative cooling can completely replace compression-based cooling systems. Please correct accordingly.

A3: We have made correction according to the Reviewer’s comments. The misleading expression has been modified as “PDRC technology is very promising to considerably decrease the use of compression-based cooling systems (e.g., air conditioners) and has a significant impact on global energy consumption”. Please see the blue words in page 3.

Q4: The authors highlight the high manufacturing cost and large-scale production limits of past works, but they do not address these topics with their own proposed emitter. How is the proposed film better than previous work regarding these aspects? What is the estimated cost? Several papers have already proposed low-cost and scalable emitters for radiative cooling, as well as companies which are already using such products. The authors should more carefully look at the existing literature’s work and appropriately acknowledge them.

A4: First, we agree with the Reviewer that several existing papers have demonstrated highly efficient, low-cost and scalable emitters for radiative cooling applications and these excellent researches also provide us valuable references. However, at present, it still remains big challenge

to develop economical materials and improve the PDRC applicability in various geographical regions and climates.

Second, in term of the cost, we selected one of the most prevalent and low-cost polymers (PMMA, ~ 3.5 \$/kg). Although the fabrication process of the PMMA_{HPA} film is not very economical (by etching in 2-5 vol% hydrofluoric acid aqueous solution), this process takes only 1 h, and each HF aqueous solution can be recycled for more than ten times. Moreover, in this study, we would just report a hierarchically structured polymer prototype with a micropore array combined with random nanopores for highly efficient day- and nighttime passive subambient radiative cooling. People may use other pore-creating methods to fabricate such structured polymer films as PDRC for practical applications. The previously reported other polymers, such as PVDF and TPX, are also conducive to radiative cooling, but they seem to be more expensive (~ 29.6 \$/kg and ~ 5.5 \$/kg, respectively). For the scalability, this work adopted the unidirectional rubbing assembly method and templating method, which are feasible for large-scale fabrication. Moreover, our PMMA_{HPA} film demonstrates excellent $\bar{\rho}_{solar}$ and $\bar{\epsilon}_{LWIR}$ and superb subambient radiative cooling performance even in hot and moist subtropical marine monsoon climate, which is not easily obtained by the reported PDRC systems.

Q5: In the introduction, the authors also suggest that their film works well in arid and humid climates as opposed to other emitters. What fundamental difference does the proposed film have with other emitters that allows it to perform better in humid conditions? One could argue that the proposed film would perform worse in humid weather conditions compared to some other selective emitters that only emit in the 8-13 μm windows and have high reflectivity outside that range. In higher humidity environment, the atmosphere is only transparent between 8-13 μm and thus any absorption outside that wavelength range decreases the cooling performance. The film proposed here has a relatively constant emissivity in the infrared, as opposed to an optimized 8-13 μm selective emissivity. Overall, we do not think the proposed film should have better performance in humid weather than previously demonstrated emitters based on its optical properties.

A5: We agree that the film would perform worse in humid weather conditions compared to some other selective emitters that only emit in the 8-13 μm windows and have high reflectivity outside that range, due to the transmissivity of the atmosphere dramatically drops in the 2nd atmospheric transparency window in humid environment. This phenomenon also exists in our test results. For example, comparing Xuzhou and Xiamen city (Figs. 6a-c and j-l), we can see that by increasing solar intensity (from ~ 860 W/m² to ~ 930 W/m²), wind speed (from ~ 0.5 m/s to ~ 1.1 m/s) and humidity (from $\sim 38\%$ to $\sim 64\%$), the subambient cooling temperatures of PMMA_{HPA} film are indeed influenced (from ~ 8.9 °C to ~ 5.5 °C). Please see the blue words in page 18.

However, in dry weather conditions, our film emitting both in the 8-13 μm and outside of the main atmospheric transparency window performs better compared to these selective emitters that

only emit in the 8-13 μm windows, because the atmospheric window is open not only in the 1st atmospheric transparency window (8-13 μm) but also in the 2nd atmospheric transparency window (16-25 μm) in dry environment.

Moreover, the fundamental differences of our PMMA_{HPA} films with other emitters are as follows: First, the hierarchical porous array on the film surface plays a crucial role in enhancing the solar reflectance and thermal emittance. The synergistic result of visible white (high solar reflectance) and infrared black (high infrared emissivity in both the 1st and 2nd atmospheric transparency window) greatly minimizes the absorbing solar irradiance and the thermal radiation emitted by the atmosphere. Second, our PMMA_{HPA} film was modified by fluorosilane to become superhydrophobic with a WCA of $\sim 156^\circ$, which is essential to ensure stable performance by restricting the effect of moisture and water under different levels of humidity. Thus, our PMMA_{HPA} films perform well even in hot and moist subtropical marine monsoon climate.

Please see the blue words in pages 4, 5 and 18.

Q6: How was the micro and nano pores size chosen? The current manuscript is missing information on the choice of these parameters and their optimization. The manuscript would strongly benefit from a theoretical model that estimates the scattering and absorption coefficients of the film based on the pore sizes and density. Such a model would also help highlight the benefits of using a hierarchical porous structure over a uniform size porous structure by showing an improvement (if it is the case) in scattering albedo in the solar spectrum when using a hierarchical porous network. The model would also give a better understanding of what is the role of the porous structure in the infrared and insights on how to improve the film further.

A6: We experimentally and theoretically demonstrated the influence of pore sizes, pore density (porosity) and structure on the optical performance and optimized these parameters accordingly.

First, we measured the reflectance spectra of the PMMA_{HPA} films with different nanopore sizes ($\sim 100, 200$ and 500 nm). As clearly seen in the added Supplementary Fig. 5a, the PMMA_{HPA} with ~ 500 nm nanopores shows a high-level $\bar{\rho}_{solar}$, but the scattering of short wavelength region is inferior to that of ~ 200 nm nanopores. The reflectance of PMMA_{HPA} with ~ 100 nm nanopores drops gradually in the visible and NIR-to-SWIR ranges (0.4 - 2.5 μm), leading to a lower $\bar{\rho}_{solar}$. Such results coincide with the added FDTD simulation data (see the added Supplementary Fig. 6a), indicating the nanopores ranging in diameter from 200 to 300 nm are optimum to reinforce the scattering of UV and visible wavelength regions.

Second, we established a periodic micropore model with different sizes in the range of 1 to 10 μm . Given the simulation results, one could derive that micropores ranging in diameter from 5 to 7 μm would highly contribute to the solar reflectance (Supplementary Fig. 6b). Thus, combining the optimized nanopore size of 200 - 300 nm and micropore size of 5 - 7 μm , the maximum $\bar{\rho}_{solar}$ of our hierarchical porous PMMA film can be obtained (Supplementary Fig. 6c).

Third, the experimental results show that solar reflectance of the PMMA_{HPA} film increases with increasing porosity, while its thermal emittance does not obviously change (Supplementary Fig. 5b). FDTD simulation results also show that the nanopore density or porosity has a positive correlation with the reflectance (Supplementary Fig. 6d). However, too high porosity may reduce the mechanical strength, making the PMMA_{HPA} film unattractive for practical uses. Based on the trade-off of the optical and mechanical properties, ~ 60 % porosity was adopted in the experiment design and radiative cooling application.

We also designed four types of PMMA models and kept their effective thickness consistent (~ 80 μm). Their simulated reflectance results are in good agreement with the experimental results (see Fig. 4e and the added Supplementary Fig. 7a), further revealing that the hierarchical porous structure containing dual-scale nano/micro cavities would significantly improve the broadband scattering performance contrast to uniform nanoscale porous structure, especially in the range of NIR-to-SWIR.

More results and discussions have been added. Please see the blue words in pages 10, 11 and 13, Supplementary pages 4-5 and the added Supplementary Figs. 5-8.

Q7: In SI Fig. 3, why are the curves for 12 μm to 100 μm thicknesses in the 4-6 μm range all in the same range of transmittance?

A7: We are very sorry for our negligence in the previous transmittance measurements. We conducted a series of transmittance measurements of PMMA_{HPA} films with various thicknesses in the range of 2.5-14 μm. It is clear from the new results that there is a general decreasing trend of transmittance with increased thickness. Please see the new Supplementary Fig. 4.

Q8: The calculation of the porosity in the SI is confusing. How do you get the thickness of the non-porous PMMA sample? Do you compress that sample until there is no more porosity or do you make another PMMA film with no porosity? Why wasn't a density-based porosity measurement used?

A8: The porosity of PMMA_{HPA} was calculated by the following equation:

$$p_{PMMA_{HPA}} \approx \frac{T_{PMMA_{HPA}} - T_{PMMA}}{T_{PMMA_{HPA}}} \times 100\% \quad (1)$$

Actually, we fabricated two types of films (PMMA_{HPA} and PMMA) with the same effective thickness by controlling the PMMA mass and film-forming substrate size during the fabrication. The only difference between the two films is the addition of pore-creating SiO₂ micro and nano spheres in PMMA_{HPA}. Please see the fabrication process in pages 22 and 23. In addition, we can see from Fig. 4d, the pristine PMMA film shows a smooth and flat surface without any pore morphology. The thickness of non-porous PMMA sample was measured at five positions *via* a

digital display micrometer with a precision of $\pm 1 \mu\text{m}$, which is equal to the effective thickness of PMMA_{HPA} film.

As the Reviewer suggested, we calculated the porosity of the PMMA_{HPA} film using a density-based porosity measurement based on the following equations:

$$P_{PMMA_{HPA}} = \frac{V_1 - V_0}{V_1} \quad (2)$$

$$V = \frac{m - m^*}{\rho_l} \quad (3)$$

Where p is the porosity of the PMMA_{HPA} film, V_1 is the volume of the porous film and V_0 is the volume of the solid film. Here the volume (V) was measured by a densimeter (ethanol was used as the liquid medium). m is the mass of the film, m^* is the mass of the film immersed in the liquid and ρ_l is the density of the liquid medium. For our PMMA_{HPA} film, when the mass ratio of SiO₂ - acetone - PMMA is 1:10:1, the calculated porosity based on the equations (2) and (3) is 61.7 %, which is almost consistent with our previous porosity calculation ($\sim 60 \%$) based on equation (1). The porosity data were obtained by three independent measurements. Thus we have re-written the porosity calculation part according to the Reviewer's suggestion. Please see the blue words in Supplementary pages 2-3.

Q9: Fig. 3b and c: Are the values of reflectance/emittance interpolated between each experimental measurement?

A9: In Fig. 3b and c, we plotted the measured original reflectance/emittance spectra of the PMMA_{HPA} films as a function of the film thickness using color filling diagram. All the values are obtained by an UV-Vis-NIR spectrophotometer and an FTIR spectrometer without any interpolation calculation. Only the emissivity was calculated as $\epsilon(\lambda) = 1 - \rho(\lambda) - \tau(\lambda)$ according to Kirchhoff's law, where $\rho(\lambda)$ and $\tau(\lambda)$ are reflectance and transmittance, respectively.

Q10: Fig. 3c, what happened at a thickness of 40 μm ? Is that an anomaly or an expected behavior?

A10: We are very sorry for our negligence in the previous emissivity measurements and have made correction for the emissivity spectra of the PMMA_{HPA} films as a function of the film thickness, especially at a thickness of 40 μm . Please see the new Fig. 3c.

Q11: Fig. 3d-f. The polarization angle is changed. Do the authors actually refer to the incident angle of the light on the sample or the actual polarization of light? The two are different and should not be confused here.

A11: To investigate the effect of ordered periodic micropore array on the absorbed infrared power, the polarization-dependent infrared emissivity spectra of our PMMA_{HPA} film at different polarization angles were characterized on an FTIR spectrometer equipped with an FT80 polarizer,

as shown in Fig. 3d-f. The average emissivity shows a regularly decreasing trend from 10° to 80°, indicating stronger infrared absorption at smaller polarization angle.

We agree with the Reviewer that the polarization angle and incident angle are two different concepts. To answer the Reviewer's questions, we further measured the angular reflectance spectra in the wavelength range of 0.4-1.1 μm at different incidence angles by an angle-resolved photonic spectral system. In the LWIR (8-13 μm) region, we simulated the average emissivity in the wavelength range of 8-13 μm along different polar angle of incidence by FDTD solutions. Please see the blue words in pages 8 and 24, Supplementary page 4 as well as the added Supplementary Fig. 2.

Q12: The authors mention that the atmospheric window emissivity is greater than 0.95 over a wide polarization angle range. Assuming they the polarization angle range refers to the incident angle, is it really good to have high emissivity at large incident angles considering that the atmospheric transmittance decreases with increasing zenith angle?

A12: With regard to the emissivity at different incidence angles, we simulated the average emissivity in the wavelength range of 8-13 μm along different polar angle of incidence by FDTD solutions. Please see the blue words in page 8, Supplementary pages 4 and the added Supplementary Fig. 2b. As the Reviewer said that the atmospheric transmittance decreases with increasing zenith angle, the simulated average emissivity of the PMMA_{HPA} film reduces to ~93 % at large emission angles. Our PMMA_{HPA} film still emits a significant part of its thermal energy at various angles through the atmospheric transparency window to the cold sink of outer space, thus is attractive for practical applications at different angles in relation to the sky.

Q13: The comparison between the PMMA_{hpa} film and the three other PMMA films is interesting, but it is not clear if the comparison is fair. Is the thickness of each film kept constant? Does the ordered micropores monolayer not achieve high solar reflectivity only because it is not thick enough or because the micropores do not have a high enough scattering coefficient? Again, a theoretical model might help here to compare the different films or pore structures/sizes for similar thicknesses and conditions.

A13: We fabricated four types of films with the same effective thickness by controlling the PMMA mass and film-forming substrate size. Please see the blue words in page 23. Thus, the comparison of optical properties for four types of PMMA films are relatively fair.

Both experimental evidence and theoretical calculations verify that the scattering of ordered micropores monolayer is weak (please see the PMMA_{MPA} curves in Fig. 4e, the added Supplementary Fig. 6b and Supplementary Fig. 7a). Moreover, we also simulated the reflectance of PMMA films with different monolayer micropore sizes (Supplementary Fig. 6b). The results indicate that only micropores monolayer do not have a high enough scattering coefficient, which is in good agreement with the experimental result. After the embedding of abundant randomized

nanopores in the monolayer micropores, the scattering efficiency dramatically increases (see PMMA_{HPA} curves in Fig. 4e and the added Supplementary Fig. 7a and Fig. 8b). We have added more discussions. Please see the blue words in pages 12-13.

Q14: What do the authors mean by “contact areas with light” at line 221?

A14: As evident from Fig. 4f that the emissivity of PMMA_{HPA} or PMMA_{NP} film is higher than that of the pristine non-porous PMMA. This probably because that the hierarchically porous structure with high specific surface area might increase the probability of infrared absorption through multiple diffuse reflection and enhances the emissivity. More importantly, the existence of abundant hierarchically pores reduce the effective refractive index and lead to a gradual refractive index transition across the polymer-air interface. Thus, the impedance matching between the porous polymer and surrounding air is improved, which reduces the surface reflectance and results in a consistently higher emissivity for PMMA_{HPA} or PMMA_{NP} film in the mid-IR wavelengths.

The previous “contact areas with light” has been revised and new literatures have been cited to support our viewpoint. Please see the blue words in pages 12-13.

Q15: An accelerated weathering test was performed for 480h. What conditions were these made in? To what equivalent real outdoor time would that correspond to?

A15: According to the Reviewer’s suggestion, the detailed weathering test has been added. The PMMA_{HPA} films were exposed in an accelerated weathering tester for 480 h undergoing harsh weathering cycles. Each cycle includes the UV irradiation at 310 nm wavelength with intensity of 0.71 W/m² at 60 °C for 4 h, followed by condensation at 50 °C for 4 h with UV lamps off. Please see the added discussions in blue words in page 15 and Supplementary page 3. Approximately, the accelerated weathering test for 480 h is equivalent to the real outdoor exposure for two years. However, it is really very hard to say this, depending upon the outdoor weather and climate.

Q16: Was there a substrate to deposit the film on during the stagnation temperature measurements? What did the setup look like during this test?

A16: The thermal box consists of insulation foam covered by a layer of reflective foil. A 10- μ m-thick transparent low-density polyethylene film was used to seal the thermal box and served as a wind shield. During the cooling temperature measurement, as shown in Fig. S1a and b below, the PMMA_{HPA} film with a size of 100 mm \times 100 mm \times 160 μ m was directly placed in the thermal box, and an adhesive resistant temperature detector was mounted on the back surface of the PMMA_{HPA} film to detect real-time temperature. During the cooling power measurement, as shown in Fig. S1c and d, the PMMA_{HPA} film was placed on a copper plate attached to a Kapton heater. This feedback-controlled heating system maintains the surface temperature of the

PMMA_{HPA} film at the measured ambient temperature to minimize the impact of conductive and convective heat losses.

Fig. S1. Schematic illustrations and photographs of cooling temperature and cooling power measurement apparatus. a, b) Thermal box apparatus for cooling temperature measurement. **c, d)** Thermal box apparatus with a feedback-controlled heater for cooling power measurement. We removed the sample and copper plate to clearly show the interior structure in **(d)**.

Q17: “This cooling performance is on par with or exceeds those in previous reports”. It would be more accurate to say that it is on par with those in previous reports as other works has demonstrated higher performance.

A17: As Reviewer suggested, it has been revised. Please see the blue words in pages 16 and 22.

Q18: Add reference for line 287-289.

A18: We have added the references 29 and 45 for line 287-289. Please see the blue words in page 17.

Q19: Line 298: The film is tested in different cities with different RH. The values however only change by 16% absolute RH. How significant is that?

A19: We specially selected three locations with different climates and investigated the effect of climate on daytime cooling capability. It is not easy to obtain high humidity under such intense sunlight (800-900 W/m²) and a clear sky.

Comparing Xuzhou and Xiamen city (Figs. 6a-c and j-l), we can conclude that increasing solar intensity (from ~ 860 W/m² to ~ 930 W/m²), wind speed (from ~ 0.5 m/s to ~ 1.1 m/s) and humidity (from ~ 38% to ~ 64%), the subambient cooling temperatures of PMMA_{HPA} film decrease from ~ 8.9 °C to ~ 5.5 °C. Thus, we can see that the humidity values change by 26% in

the two locations and this much difference is of great significance under such intense sunlight (800-900 W/m²). More discussions have been added. Please see the blue words in page 18.

Q20: Fig. 8: What atmospheric model was used for the cooling power curves? The high nighttime cooling power of nearly 140 W/m² suggests that a very low humidity environment was used for the modeling, which might differ with the experimental locations.

A20: Thank you so much for your kind suggestions. Atmospheric conditions, such as humidity and cloud cover, can considerably affect the cooling power results. Here, the atmospheric transmittance was evaluated with MODTRAN6 according to the references below. We considered the atmosphere as a unity and used the ambient temperature to replace the practical temperature which varied with altitude.

As the Reviewer said, the atmospheric model we used in the calculations might not match the experimental locations well (Shanghai city, April 23, ~ 44% RH at noon), but we can approximately compare the experimental and theoretical results *via* this model. In addition, the main reason of the high nighttime cooling power is superior thermal emittance ($\bar{\epsilon}_{LWIR} = 0.98$) of the PMMA_{HPA} film. We have added the references 34 and 51 in page 25.

1. Berk, A. *et al.* MODTRAN6: a major upgrade of the MODTRAN radiative transfer code. *Proc. SPIE 9088, Algorithms and Technologies for Multispectral, Hyperspectral, and Ultraspectral Imagery XX*, 90880H (2014); doi: 10.1117/12.2050433.

2. Leroy, A. *et al.* High-performance subambient radiative cooling enabled by optically selective and thermally insulating polyethylene aerogel. *Sci. Adv.* **5**, eaat9480 (2019).

To Reviewer #3:

General comments: Passive radiative cooling with no energy input is an appealing technology to meet the demand of cooling. Therefore, recently people are developing all kinds of materials or structures to enhance the cooling performance. The authors proposed a polymer (i.e., PMMA) with a hierarchically porous array. It can achieve high solar reflectance and thermal emittance, and therefore, could be employed to realize passive daytime radiative cooling. This manuscript should be strengthened with more clarification, experimental results and further analysis on the following details.

General answers: Thanks the Reviewer very much for his or her positive comments and valuable suggestions. We are trying our best to revise our manuscript based on these comments and suggestions.

Q1: The structural polymer proposed by the authors is also one kind of porous polymer mentioned in the introduction of the manuscript. What are the advantages of the proposed structural polymer compared with the ones reported previously?

A1: There are some significant differences between our structural PMMA and the previously

reported porous polymers mentioned in the introduction. The main features and innovations of our work are as follows:

(1) Our PMMA_{HPA} film has a hierarchically structure with a close-packed micropore array on the surface combined with abundant random nanopores by a templating method, which is reported for the first time. Importantly, these micropores and nanopore and their arrangements can be precisely controlled by templating colloidal spheres. Thus, we can further compare the optical properties of our hierarchically porous array PMMA (PMMA_{HPA}) film with three other types of PMMA films (*e.g.*, nanopore PMMA_{NP}, monolayer micropore array PMMA_{MPA} and pristine PMMA). Both experimental evidence and theoretical calculations verify the hierarchical porous structure with a micropore array on the film surface combined with abundant randomized nanopores plays crucial roles in enhancing the solar reflectance. This study and hierarchically porous structured PMMA prototype may provide deep insight into the crucial roles of micropores and nanopores in solar reflectance and thermal emittance and help us design and fabricate more efficient all-day passive subambient radiative cooling materials and systems.

(2) Because of the unique structure and favorable intrinsic optical properties, the as-obtained PMMA_{HPA} film demonstrates excellent $\bar{\rho}_{solar}$ and $\bar{\epsilon}_{LWIR}$ and superb subambient radiative cooling performance in various geographical regions and climates.

(3) As one of the most widely used and low-cost polymers, PMMA is chosen as our cooling radiator (~3.5 \$/kg). Pristine PMMA film is highly transparent with negligible extinction coefficient in the solar wavelengths as well as multiple extinction peaks within the LWIR window, which is responsible for outstanding PDRC performance. The previously reported polymers, such as PVDF and TPX, are also conducive to radiative cooling, but they seem to be more expensive (~29.6 \$/kg and ~5.5 \$/kg, respectively).

We hope our study can help people to further deep understand the roles of micropores and nanopores and their arrangement in optical properties and to find better PDRC materials and systems for practical use.

Q2: The authors mentioned the surface modification by fluorosilane, but did not show the detailed experiment process. It is also important to compare the reflectance, emissivity and the radiative cooling performance before and after the surface modification.

A2: According to the Reviewer's suggestions, we have added the experiment process of the surface modification by fluorosilane. Please see the blue words in page 23. We also evaluated the optical performance of the PMMA_{HPA} film before and after the surface modification, both the solar reflectance and infrared emissivity are at high levels and vary negligibly after fluorosilane treatment. Please see the blue words in page 15 and the added Supplementary Fig. 11.

We further investigated the effect of surface modification on the daytime radiative cooling performance of the PMMA_{HPA} film on August 19, 2020, under a solar intensity of ~ 800 W/m² and a relative humidity of ~ 51% in Shanghai city. The results show that the PMMA_{HPA} films with

and without surface modification can achieve subambient cooling temperatures of ~ 6.9 °C and ~ 6.5 °C, respectively. While the PMMA_{HPA} film without surface modification also demonstrates excellent passive radiative cooling behavior, the fluorosilane treatment is required to ensure the stable performance and improve the durability by restricting the effect of moisture and water under different levels of humidity. Please see the blue words in page 18 and the added Supplementary Fig. 14.

Q3: The details of the weathering test are not shown, including how heat, water, oxygen and UV radiation were applied on the polymer.

A3: According to the Reviewer's suggestion, we have showed the details of the weathering test. The PMMA_{HPA} films were exposed in an accelerated weathering tester for 480 h undergoing harsh weathering cycles. Each cycle includes the UV irradiation at 310 nm wavelength with intensity of 0.71 W/m^2 at 60 °C for 4 h, followed by condensation at 50 °C for 4 h with UV lamps off. Please see the added discussions in blue words in page 15 and Supplementary page 3.

REVIEWER COMMENTS

Reviewer #1 (Remarks to the Author):

I appreciate the authors' revision, and I think the quality of the paper can be improved to be qualified for nature comm, but during my reading, I find out more issues in the manuscript, including various descriptions and results that may be scientifically incorrect. I would like to point them out here and look forward to the authors' response. These issues need to be clearly addressed before acceptance.

1. The mixed use of "effective thickness" and "thickness". Effective thickness means the equivalent thickness of PMMA if the film is solid according to the authors, as supported by statements in SI:

(~ 160 μm effective thickness was used in experiments in Fig. 4)

It is noteworthy that all the effective thicknesses in this article are 40% of the measured thicknesses due to the ~ 60% porosity of the PMMAHPA film.

This is fine. However, in various places, only "thickness" is used, such as caption of figure 3 and the main text. This is very misleading, because readers will think that the thickness (including pores) is 160 μm . This issue appears through the whole paper, which should be corrected.

2. In all previous studies, the angle-averaged emittance $\bar{\epsilon}$ is the average over **incident angles**, but not **polarization angles**. The authors state that it is over polarization angles in Figure 3 caption and manuscript (e.g. line 221 on page 11, line 452 on page 24).

Moreover, On line 57 in SI, it is mentioned that "in the LWIR (8-13 μm) region, we simulated the average emissivity along different polar angle of incidence." But in figure S2, it says different incident angles. The use of terms are confusing and not consistent. This must be corrected, and emittance is based on incident angle-averaged.

3. More importantly, the reflectance in thermal radiation wavelengths is measured without an integrated sphere based on description in methods and information on Nicolet 6700 IR spectrophotometer. So if for the mirror reflection angle, 2% of light is received, then the overall reflection could be much higher (e.g.10%). The emittance = 1- reflectance will be much smaller. The emittance claimed is very doubtful.
4. The simulation results are weird. With the open half spherical structure near surface, the electrical field should show boundary of the porous polymer. I have performed the same simulation, and both small and large pores can be seen. Has authors used the correct mesh size? Size of $\sim 1/10$ of wavelength or less should be used.

Simulation of the same structure for wavelength = 0.5 ,1 , and 2 um by COMSOL

[Redacted]

And also from other research (Science Advances 08 Apr 2016: Vol. 2, no. 4, e1501227 DOI: 10.1126/sciadv.1501227)

5. As PE cover is used in all measurements in Fig. 6 and 7 (at least the paper does not mention a case that PE is removed), the sunlight that the film received should be 10-30% less than the pyranometer outside the box. This should be mentioned or calibrated.
6. Where is ambient temperature measured. Inside the box or outside the box? If inside, only “cooler than air temperature near the sample” can be claimed instead of “cooler than ambient temperature”. The difference is whether the sample receives energy from the atmosphere or not. Because ambient means the temperature of atmosphere, and the temperature of air is typically higher than the ambient. If the temperature is in the white thermostat box on the left side in Fig. 7c, it is also not correct, because 1) The box is sealed without ventilation. 2) It is on ground, which can be heated.
7. I suggest authors report the local temperature by meteorological station, such as <https://www.wunderground.com/history/daily/cn/xuzhou/ZSXZ/date/2020-5-2>

8. A minor point. The schematic in Fig. 1 appears that the film is only ~ 10 μm thick. It is better to make the film thicker.

Reviewer #2 (Remarks to the Author):

We believe the authors have done a good job at addressing the technical comments which has improved the quality of the work. While the technical content of the work has improved, the novelty and impact of the work is still limited and unclear compared to previous work in the literature (such as porous hierarchical polymer structure for radiative cooling in Science DOI: 10.1126/science.aat9513). As such, we believe the paper is better suited for publication in another journal than Nature Communications. Below are comments and suggestions which we hope will help further improve the quality of the paper.

1. While we understand the novelty in the design and fabrication of the hierarchically porous polymer, the impact of the work on the radiative cooling community is still unclear. The authors should make it clearer how their selective emitter design enables them to achieve features or performance not achievable with previous work. To our understanding, the selective emitter proposed here achieves very good solar reflectance and infrared emittance, but its performance, cost and scalability are similar to some of the past work.
2. Fig. 2 a,b: Why is there a sudden vertical change in reflectance and emittance at 2.5 μm (which also coincides with change of detector and light source)? According to the refractive index (n,k) and the feature size in this work, we should expect a more gradual change?
3. Fig. 2 f: The schematic might not be an accurate representation of what is actually happening with the light-matter interaction. The way it is currently drawn suggests specular reflections and surface reflections only while it seems like practically, we would get diffusion reflections and a volumetric scattering of light by the medium. Does that affect your interpretation of the enhanced reflection from the hierarchical structure?
4. It is still unclear what atmospheric spectrum the authors used for Fig. 8 and still seems way too optimistic to find in typical regions around the world (especially humid regions where this work seems to put emphasis). The authors replied to our previous comment:
"Thank you so much for your kind suggestions. Atmospheric conditions, such as humidity and cloud cover, can considerably affect the cooling power results. Here, the atmospheric transmittance was evaluated with MODTRAN6 according to the references below. We considered the atmosphere as a unity and used the ambient temperature to replace the practical temperature which varied with altitude."
It is not clear what it meant by "considered the atmosphere as a unity and used the ambient temperature to replace the practical temperature which varied with altitude". The authors should be careful in modifying any atmospheric model so that they remain realistic. Also, in the reference highlighted by the authors, they mention that they used the US Standard 1976 spectrum for the atmosphere which doesn't seem to be the case here.

Reviewer #3 (Remarks to the Author):

The revised manuscript better explained the difference between their proposed structure and the ones reported previously. Besides, the details of the experiments were added.

I recommend this manuscript to be published.

To Reviewer #1:

General comments: I appreciate the authors' revision, and I think the quality of the paper can be improved to be qualified for nature comm, but during my reading, I find out more issues in the manuscript, including various descriptions and results that may be scientifically incorrect. I would like to point them out here and look forward to the authors' response. These issues need to be clearly addressed before acceptance.

General answers: Thanks the Reviewer very much for his or her positive comments and highly valuable suggestions. We are trying our best to revise our manuscript according to these comments and suggestions.

Q1: The mixed use of “effective thickness” and “thickness”. Effective thickness means the equivalent thickness of PMMA if the film is solid according to the authors, as supported by statements in SI: (~ 160 μm effective thickness was used in experiments in Fig. 4).

It is noteworthy that all the effective thicknesses in this article are 40% of the measured thicknesses due to the ~ 60% porosity of the PMMA_{HPA} film.

This is fine. However, in various places, only “thickness” is used, such as caption of figure 3 and the main text. This is very misleading, because readers will think that the thickness (including pores) is 160 μm . This issue appears through the whole paper, which should be corrected.

A1: We are very sorry for our negligence in the thickness descriptions and have made correction for the whole paper. Please see the blue words in pages 7, 9, 10, 11, 23 and Supplementary page 9.

Q2: In all previous studies, the angle-averaged emittance $\bar{\epsilon}$ is the average over incident angles, but not polarization angles. The authors state that it is over polarization angles in Figure 3 caption and manuscript (e.g. line 221 on page 11, line 452 on page 24).

Moreover, on line 57 in SI, it is mentioned that “in the LWIR (8-13 μm) region, we simulated the average emissivity along different polar angle of incidence.” But in figure S2, it says different incident angles. The use of terms are confusing and not consistent. This must be corrected, and emittance is based on incident angle-averaged.

A2: We agree with the Reviewer that the polarization angle and incident angle are two different concepts. Actually, we demonstrated both the polarization angle and incident angle in our studies. First, we measured the polarization-dependent infrared emissivity spectra of our PMMA_{HPA} film at different polarization angles using an FTIR spectrometer equipped with a polarizer, as shown in Figs. 3d-f. We have revised the caption for Fig. 3e to make it clear, please see the blue words in page 11.

In addition, we simulated the average emissivity in the wavelength range of 8-13 μm at different incidence angles by FDTD solutions, as shown in Supplementary Fig. 2b. The misleading expression of “different polar angle of incidence” in Supplementary page 4 has been

revised to “different incidence angles”.

Q3: More importantly, the reflectance in thermal radiation wavelengths is measured without an integrated sphere based on description in methods and information on Nicolet 6700 IR spectrophotometer. So if for the mirror reflection angle, 2% of light is received, then the overall reflection could be much higher (e.g.10%). The emittance = 1- reflectance will be much smaller. The emittance claimed is very doubtful.

A3: The reflectance in the mid-infrared (2.5 - 25 μm) wavelength ranges was characterized in an FTIR spectrometer equipped with a Smart Diffuse Reflectance accessory. An integral gold mirror was used as a background reference.

For mid-infrared diffuse analysis, the Thermo Scientific Smart Diffuse Reflectance accessory is highly effective at maximizing diffusely scattered radiation while minimizing specular reflected radiation which is a source of spectral interference. As shown in Fig. R1 below, the device uses ellipsoidal collecting mirrors and compound parabolic concentrator (CPC), which are efficient in terms of collecting and transmitting diffusely reflected radiation to the detector. So, the reflection of the samples in the range of 2.5-25 μm is lower, resulting in a higher emittance even for the pristine PMMA film (Fig. 4f in page 14).

Fig. R1. Thermo Scientific Smart Diffuse Reflectance accessory. (a) Optical diagram of the diffuse reflection infrared spectrometry. (b) Photo of the Smart Diffuse Reflectance accessory of Nicolet 6700, Thermo Fisher Scientific, USA. (c) Compound Parabolic concentrator (CPC) optics.

Q4: The simulation results are weird. With the open half spherical structure near surface, the electrical field should show boundary of the porous polymer. I have performed the same simulation, and both small and large pores can be seen. Has authors used the correct mesh size? Size of $\sim 1/10$ of wavelength or less should be used. And also from other research (Science Advances 08 Apr 2016: Vol. 2, no. 4, e1501227 DOI: 10.1126/sciadv.1501227)

A4: Thanks for the kind suggestion. The electric field figures in our previous version are anamorphic and compressed. We re-exported the new electric field figures of four types of PMMA films at wavelength of $\sim 2 \mu\text{m}$ by FDTD Solutions. Please see the new Supplementary Fig. 8a. The revised version clearly shows the boundary of the porous polymer and is similar to

the Reviewer's simulation results by COMSOL. The paper the Reviewer mentioned has been consulted and cited as reference 45.

Owing to the computational limits and overlong computing time, the mesh size we used for the FDTD simulations is 50 nm for thick PMMA ($\sim 1/6$ of the shortest wavelength) and 10 nm for thin PMMA ($\sim 1/30$ of the shortest wavelength). Please see the the blue words in Supplementary page 4. Here, we also performed the simulations of the thin PMMA_{HPA} film with different mesh size (10, 30 and 50 nm) by FDTD Solutions, as shown in Fig. R2 below. Apparently, the electric field distributions for wavelength of ~ 0.5 , 1.0, 1.5 and 2.0 μm show that the smaller the mesh size is, the clearer the boundary is. But the mesh size of 10 nm is rarely feasible for thicker PMMA due to the computational limits. In addition, the simulated reflectance and emittance results of PMMA_{HPA} film using smaller mesh size (*e.g.*, 10 nm) show negligible variations comparing with larger mesh size (*e.g.*, 30 and 50 nm).

Fig. R2. Electric field distributions of the PMMA_{HPA} film for wavelength of ~ 0.5 , 1.0, 1.5 and 2.0 μm by FDTD Solutions. (a) Mesh size of 10 nm. (b) Mesh size of 30 nm. (c) Mesh size of 50 nm.

Q5: As PE cover is used in all measurements in Fig. 6 and 7 (at least the paper does not mention a case that PE is removed), the sunlight that the film received should be 10-30% less than the pyranometer outside the box. This should be mentioned or calibrated.

A5: According to the Reviewer's suggestion, we have mentioned that the sunlight our PMMA films received is less than the pyranometer measured due to the PE cover on the thermal box. In addition, we measured the spectral transmittance of the 10- μm -thick low-density polyethylene (LDPE) film that used in the box, showing a high transmittance ($\sim 90\%$) of the LDPE film in the solar wavelengths (Fig. R3 below). So approximately $\sim 10\%$ sunlight is blocked by the LDPE film

during the PDRC measurements. More details have been added, please see the blue words in page 27.

Fig. R3. Spectral transmittance of the 10- μm -thick LDPE film in the range of solar spectrum from 0.3 to 2.5 μm .

Q6: Where is ambient temperature measured. Inside the box or outside the box? If inside, only “cooler than air temperature near the sample” can be claimed instead of “cooler than ambient temperature”. The difference is whether the sample receives energy from the atmosphere or not. Because ambient means the temperature of atmosphere, and the temperature of air is typically higher than the ambient. If the temperature is in the white thermostat box on the left side in Fig. 7c, it is also not correct, because 1) The box is sealed without ventilation. 2) It is on ground, which can be heated.

A6: Thanks for the kind suggestion. During our radiative cooling measurements, a temperature detector was mounted outside the box to detect real-time temperature of the ambient. More details have been added, please see the blue words in page 26.

Q7: I suggest authors report the local temperature by meteorological station, such as <https://www.wunderground.com/history/daily/cn/xuzhou/ZSXZ/date/2020-5-2>

A7: Thanks for the Reviewer’s suggestion, we have reported the local temperature by meteorological station of the three cities and different dates. Please see the blue words in Supplementary page 18.

Q8: A minor point. The schematic in Fig. 1 appears that the film is only ~ 10 μm thick. It is better to make the film thicker.

A8: According to the Reviewer’s suggestion, we have made the schematic film thicker. Please see the new Fig. 1a.

To Reviewer #2:

General comments: We believe the authors have done a good job at addressing the technical comments which has improved the quality of the work. While the technical content of the work has improved, the novelty and impact of the work is still limited and unclear compared to previous work

in the literature (such as porous hierarchical polymer structure for radiative cooling in Science DOI: 10.1126/science.aat9513). As such, we believe the paper is better suited for publication in another journal than Nature Communications. Below are comments and suggestions which we hope will help further improve the quality of the paper.

General answers: Thanks the Reviewer very much for his or her constructive comments.

We acknowledge that our idea to design this hierarchically structured polymer film is inspired by the previously excellent works including Science DOI: 10.1126/science.aat9513. However, there are some significant differences between our work and the previous works as follows, which may find the main innovations and impacts of our manuscript:

(1) Our PMMA_{HPA} film has a hierarchically structure with a close-packed micropore array on the surface combined with abundant random nanopores and this unique structure is reported for the first time. Both experimental evidence and theoretical calculations confirm that the hierarchically porous structure with a micropore array on the film surface combined with abundant randomized nanopores can efficiently enhance the solar reflectance and thermal emittance.

(2) Because of the unique structure above and favorable intrinsic optical properties of the widely used and low-cost PMMA, the as-obtained PMMA_{HPA} film demonstrates excellent $\bar{\rho}_{solar}$ and $\bar{\epsilon}_{LWIR}$ without needing any silver or aluminum reflectors, realizing excellent superb subambient radiative cooling performance in various geographical regions and climates compared to the previously reported systems.

(3) This study and the structured PMMA prototype provide deep insight into the effects of micropores, nanopores and their arrangement on optical performance and help us design and fabricate more efficient all-day passive and all-climate subambient radiative cooling materials and systems for practical uses.

Q1: While we understand the novelty in the design and fabrication of the hierarchically porous polymer, the impact of the work on the radiative cooling community is still unclear. The authors should make it clearer how their selective emitter design enables them to achieve features or performance not achievable with previous work. To our understanding, the selective emitter proposed here achieves very good solar reflectance and infrared emittance, but its performance, cost and scalability are similar to some of the past work.

A1: The novelty and impact of this work can be described as follows:

Theoretically, our study and the unique structure may provide deep insight into the effects of micropores, nanopores and their arrangement on optical performance that can help us design and fabricate more efficient all-day and all-climate passive subambient radiative cooling materials and systems for practical uses. The hierarchical porous array on the film surface plays a crucial role in enhancing the solar reflectance and thermal emittance. The synergistic result of visible white (high solar reflectance) and infrared black (high infrared emissivity in both the 1st and 2nd atmospheric transparency window) greatly minimizes the absorbing solar irradiance and the thermal radiation

emitted by the atmosphere.

Technically, our study provides a new prototype with a close-packed micropore array on the surface combined with abundant random nanopores inside, which can exhibit superior cooling performances, including both usual weather and sticky subtropical marine monsoon climate, this is all-day and all-climate, which was not reported in previous studies. And the superhydrophobized PMMAHPA film can ensure cooling performance durability by eliminating the effect of moisture and water under different levels of humidity. This prototype can tell people design and fabricate highly efficient PDRC systems for practical applications in various climates. In addition, PMMA (~3.5 USD /kg) is remarkably cheaper than the previously reported polymer emitters, such as PVDF and TPX (~29.6 USD/kg and ~5.5 USD/kg, respectively).

To highlight the novelty and impact of this work compared to the previously reported works, more descriptions have been added, please see the blue words in pages 5 and 22.

Q2: Fig. 2 a,b: Why is there a sudden vertical change in reflectance and emittance at 2.5 μm (which also coincides with change of detector and light source)? According to the refractive index (n,k) and the feature size in this work, we should expect a more gradual change?

A2: The spectral reflectance of the films was determined separately in ultraviolet, visible and near-infrared (0.3 - 2.5 μm) range and mid-infrared (2.5 - 25 μm) range with two spectrometers. The 2.5 μm wavelength is exactly the cut-off point of the two spectrometers and a sudden vertical change in reflectance and emittance at 2.5 μm might be due to the change of detector and light source. For the ultraviolet, visible and near-infrared (0.3 - 2.5 μm) wavelength ranges, measurements were taken using an UV-Vis-NIR spectrophotometer equipped with a deuterium lamp for UV region, tungsten-halogen lamp for Vis, NIR range and a polytetrafluoroethylene integrating sphere. For the mid-infrared (2.5 - 25 μm) wavelength ranges, measurements were taken using an FTIR spectrometer equipped with a mercury cadmium telluride detector and a Smart Diffuse Reflectance accessory. An integral gold mirror is used as a background reference.

According to the Reviewer's suggestion, we plotted the spectral reflectance of the PMMA_{HPA} film from 0.3 to 25 μm again. The new version shows negligible variations in reflectance and emittance compared with previous version ($\bar{\rho}_{solar} = 0.95$, $\bar{\epsilon}_{LWIR} = 0.98$), but a more gradual change can be seen at 2.5 μm . Please see the new Figs. 2 a and b.

Q3: Fig. 2 f: The schematic might not be an accurate representation of what is actually happening with the light-matter interaction. The way it is currently drawn suggests specular reflections and surface reflections only while it seems like practically, we would get diffusion reflections and a volumetric scattering of light by the medium. Does that affect your interpretation of the enhanced reflection from the hierarchical structure?

A3: Thank you so much for your kind suggestion. The schematic diagram in Fig. 2f has been revised. The current diagram shows multiple reflections including specular reflections, diffusion

reflections and volumetric scattering, which is better to explain the enhanced reflection from the hierarchical structure in our manuscript. Please see the new Fig. 2f.

Q4: It is still unclear what atmospheric spectrum the authors used for Fig. 8 and still seems way too optimistic to find in typical regions around the world (especially humid regions where this work seems to put emphasis). The authors replied to our previous comment:

“Thank you so much for your kind suggestions. Atmospheric conditions, such as humidity and cloud cover, can considerably affect the cooling power results. Here, the atmospheric transmittance was evaluated with MODTRAN6 according to the references below. We considered the atmosphere as a unity and used the ambient temperature to replace the practical temperature which varied with altitude.”

It is not clear what it meant by “considered the atmosphere as a unity and used the ambient temperature to replace the practical temperature which varied with altitude”. The authors should be careful in modifying any atmospheric model so that they remain realistic. Also, in the reference highlighted by the authors, they mention that they used the US Standard 1976 spectrum for the atmosphere which doesn't seem to be the case here.

A4: We are very sorry for our negligence in previous net cooling power calculations. According to the Reviewer's suggestion, we used the Mid-Latitude Summer Atmosphere Model ([http://modtran.spectral.com/modtran home](http://modtran.spectral.com/modtran_home)) to calculate the net cooling power (P_{cool}), which matches the experimental locations well (Shanghai city, 31° 18' 22" N, 121° 30' 17" E).

A maximum cooling power of 124.40 W/m² and 74.40 W/m² can be achieved for nighttime and daytime operations, respectively. These new calculated net cooling power results are slightly lower than previous calculations due to the atmospheric transmittance difference between Mid-Latitude Summer and US Standard 1976. We have made correction, please see the blue words in page 21, 25 and the new Fig. 8.

To Reviewer #3 (Remarks to the Author):

General comments: The revised manuscript better explained the difference between their proposed structure and the ones reported previously. Besides, the details of the experiments were added. I recommend this manuscript to be published.

General answers: Thanks the Reviewer very much for his or her positive comments.

REVIEWER COMMENTS

Reviewer #1 (Remarks to the Author):

I think the revision does not directly answer some of my questions. I can give the authors another chance to revise, but it is important that the authors do the right things.

1. The configuration in Figure R1 cannot be used instead of an integrated sphere. As shown in the following figure, scattered light with red arrows will not be collected, so the emissivity is overestimated. I agree that such porous polymer film should have a high emissivity, but correct measurement must be done to report correct values.

This doubt also echoes with the question 2 by reviewer 2. Because no integrated sphere is used and some scattered light is not collected, so emissivity is over-estimated, which explains the jump at the wavelength of 2.5 μm .

2. In figure S2, the first sentence is still on polar distribution. It should be incident angle. In a, Is it 0.4-2.5 μm or 0.4-1.1 μm ? If it is 0.4-1.1, the authors should not claim solar reflectance. Please pay attention to such things. They are important.

3. The authors don't give mathematical definition of solar reflectance and thermal emissivity. This should be given.

Reviewer #2 (Remarks to the Author):

We thank the authors for addressing our previous comments. While we believe the technical content of the work has improved again with this new revision and that the fabrication process and hierarchical structure are impressive, the novelty and impact of the work in the field of radiative cooling is still limited and unconvincing. We are not convinced about how this new selective emitter will enable better performance and cheaper radiative cooling systems. As such, we believe the paper is better suited for publication in another journal than Nature Communications. Below are further comments and suggestions which we hope will help further improve the quality of the paper.

1. Based on the transmittance characterization in the supplementary figure 4, there is around 4-5% transmittance in the solar spectrum for the thickest sample. However, it seems like the sample was directly mounted on a copper surface for the outdoor experiment and for some of the optical measurement in the manuscript, which may enhance the overall solar reflectance during onsite characterization. The authors should make it clear which system (film only or film on copper) is used when measuring the optical properties and when performing the outdoor experiments. The authors should also make sure to show the optical properties of their assembly emitter (film on substrate) that was used during the outdoor experiments. How is the performance of the radiative cooler affected by the substrate optical properties in that case?
2. Figure 4e indicates the solar reflectance of PMMA NP as 0.74, which is rather poor for radiative cooling. Some of commercial white paint can achieve 85%. This is fine since the actual sample is PMMA HPA. However, this is not consistent with the results in Figure 5d that the PMMA NP almost achieved daytime cooling as its temperature was very similar to the ambient temperature. This should not be the case for a coating of 74% solar reflectance. It could be possible that the copper surface helps a lot in the outdoor tests such that the real solar reflectance is higher. Again, the authors should clearly indicate and differentiate what are the optical properties of the film alone and the film plus substrate (copper).
3. The simulation of the solar reflectance in Figure S7 is lower than the experimental characterization in Figure 2a. Usually the theoretical estimation is similar or higher than experimental results due to imperfection in fabrication process. Could the authors comment on the possible causes?
4. The authors claim that the previous approaches with silver mirror "appears to be costly". However, the fabrication method in this work involves SiO₂ microspheres with narrow particle size range and HF etching. As such, it is difficult to agree with the authors' claim that their film is cheaper simply based on raw material cost due to the high possible manufacturing process costs. The porous polymer or recent progress of cooling paints can be packaged into cans and sprayed rather conveniently, and silver layer can be rolled easily as well, which could also translate into cheaper installations costs than the current approach.

Point-by-Point to Reviewers' Comments

To Reviewer #1:

General comments: I think the revision does not directly answer some of my questions. I can give the authors another chance to revise, but it is important that the authors do the right things.

General answers: Thanks the Reviewer very much for his or her highly valuable suggestions. Now we believe that we have answered all the questions by doing our best.

Q1: The configuration in Figure R1 cannot be used instead of an integrated sphere. As shown in the following figure, scattered light with red arrows will not be collected, so the emissivity is overestimated. I agree that such porous polymer film should have a high emissivity, but correct measurement must be done to report correct values.

This doubt also echoes with the question 2 by reviewer 2. Because no integrated sphere is used and some scattered light is not collected, so emissivity is over-estimated, which explains the jump at the wavelength of 2.5 μm .

A1: According to the Reviewer's valuable suggestions, we have measured the reflectance in the wavelength range of 2.5-16 μm in an FTIR spectrometer (Nicolet 6700, Thermo Fisher Scientific, USA) with a gold integrating sphere. The emissivity was calculated as $\varepsilon(\lambda) = 1 - \rho(\lambda) - \tau(\lambda)$.

Compared with previous emittance results measured with smart diffuse reflectance accessory, the new emittance spectrum of our PMMA_{HPA} film ($\sim 160 \mu\text{m}$ effective thickness) shows a more gradual change at 2.5 μm and drops slightly in the range of 2.5-6 μm . Usually, this range of emittance spectrum (2.5-6 μm) was not concerned in the average emittance calculation by the previous studies and the average emittance ($\bar{\varepsilon}_{LWIR}$) in the LWIR atmospheric transmittance window is defined as:

$$\bar{\varepsilon}_{LWIR} = \frac{\int_{8\mu\text{m}}^{13\mu\text{m}} I_{BB}(\lambda) \varepsilon_{LWIR}(\lambda, \theta) d\lambda}{\int_{8\mu\text{m}}^{13\mu\text{m}} I_{BB}(\lambda) d\lambda} \quad (1)$$

where $I_{BB}(\lambda)$ is the spectral intensity emitted by a blackbody and $\varepsilon_{LWIR}(\lambda, \theta)$ is the surface's angular spectral thermal emittance in the range of 8 - 13 μm .

However, the average emittance of our PMMA_{HPA} film in the range of 8-13 μm shows negligible variation ($\bar{\varepsilon}_{LWIR} = 0.975 \pm 0.01$ by integrating sphere compared to 0.980 ± 0.01 by smart diffuse reflectance). Since all the previously reported average emittance was accurate to the second decimal point, 0.98 is used in our manuscript for sake of comparison.

In addition, three other types of PMMA films, PMMA_{HPA} films with different thickness and porosity, loose-packed PMMA_{HP} film, PMMA_{HPA} films before and after the surface modification, and PMMA_{HPA} films before and after UV accelerated weathering treatment have been also

measured again with a gold integrating sphere. The new results can be seen in the new Figs. 2a and b, Figs. 3a and c, Fig. 4f, Supplementary Figs. 9e, 11b and 12f. Given the slight variations in the range of 2.5-6 μm and negligible variations in the range of 8-13 μm of all the samples, we have made correction accordingly in the description of our manuscript. Please see the blue words in pages 10, 12 and 23, and the updated Figures.

Q2: In figure S2, the first sentence is still on polar distribution. It should be incident angle. In a, is it 0.4-2.5 μm or 0.4-1.1 μm ? If it is 0.4-1.1, the authors should not claim solar reflectance. Please pay attention to such things. They are important.

A2: We are very sorry for our negligence and have revised the caption for Supplementary Figure 2, please see the blue words in Supplementary page 7.

Q3: The authors don't give mathematical definition of solar reflectance and thermal emissivity. This should be given.

A3: The mathematical definition of solar reflectance and thermal emissivity were given in our manuscript. Please see the equations (6) and (7) in pages 25-26.

To Reviewer #2:

General comments: We thank the authors for addressing our previous comments. While we believe the technical content of the work has improved again with this new revision and that the fabrication process and hierarchical structure are impressive, the novelty and impact of the work in the field of radiative cooling is still limited and unconvincing. We are not convinced about how this new selective emitter will enable better performance and cheaper radiative cooling systems. As such, we believe the paper is better suited for publication in another journal than Nature Communications. Below are further comments and suggestions which we hope will help further improve the quality of the paper.

General answers: Thank the Reviewer very much for his or her encouraging comments and concern.

First, we acknowledge that many outstanding materials and structures for passive radiative cooling have been published recently and we have added the references 18, 35, 38 and 39 in the Introduction part to better review recent progress in the radiative cooling field. In addition, we have made corrections for the inappropriate descriptions in the Introduction. Please see the blue words in page 4.

Second, the novelty and impact of this work in the field of radiative cooling can be described as follows:

Theoretically, instead of using a reflective metallic mirror or a multilayer photonic structure and randomly distributed porous polymer films to reject the full spectrum solar radiation, our

study has demonstrated the effects of micropores, nanopores and their arrangement on optical performance for the first time, which may provide deep insight into the effects of various pores and their array and help us design and fabricate more efficient all-day passive subambient radiative cooling materials and systems for practical uses. The hierarchical porous array on the film surface plays a crucial role in enhancing the solar reflectance and thermal emittance.

Technically, our study provides a new prototype with a close-packed micropore array on the surface combined with abundant random nanopores inside, which can exhibit superior cooling performances, including $\bar{\rho}_{solar}$ (0.95) and $\bar{\epsilon}_{LWIR}$ (0.98), with subambient cooling of as high as ~ 8.2 °C during the night and of ~ 6.0 °C to ~ 8.9 °C during midday with an average cooling power of ~ 85 W/m² under solar intensity of ~ 900 W/m², and promisingly ~ 5.5 °C even under solar intensity of ~ 930 W/m² and relative humidity of $\sim 64\%$ in hot and moist subtropical marine monsoon climate, this is all-day and all-climate, which was not reported in previous studies. Moreover, our PMMA_{HPA} film was modified by fluorosilane to become superhydrophobic with a WCA of $\sim 156^\circ$, which is essential to ensure stable performance and improve the durability by restricting the effect of moisture and water under different levels of humidity. This prototype can tell people to design and fabricate highly efficient PDRC systems for practical applications in various climates.

To highlight this novelty, some revisions have been made, please see the line 20 in page 5 and the blue words in page 22.

Q1: Based on the transmittance characterization in the supplementary figure 4, there is around 4-5% transmittance in the solar spectrum for the thickest sample. However, it seems like the sample was directly mounted on a copper surface for the outdoor experiment and for some of the optical measurement in the manuscript, which may enhance the overall solar reflectance during onsite characterization. The authors should make it clear which system (film only or film on copper) is used when measuring the optical properties and when performing the outdoor experiments. The authors should also make sure to show the optical properties of their assembly emitter (film on substrate) that was used during the outdoor experiments. How is the performance of the radiative cooler affected by the substrate optical properties in that case?

A1: First, during the spectral reflectance measurement, we put one black substrate behind the samples to eliminate the reflectance contribution of the substrate. Please see the Characterization part in pages 23-24.

Second, as shown in Fig. R1 below, the copper plate attached to Kapton heater was only used during the radiative cooling power measurement to better feedback-control the film at ambient temperature and accurately assess the cooling power. But, during the outdoor cooling temperature measurement, the heater and copper plate were removed and an adhesive resistant temperature detector was directly mounted on the back surface of the polymer film to detect real-time temperature of the samples. Please see the measurement details in pages 26 and 27.

Therefore, in both optical and cooling measurements, we have avoided the effect of any substrate.

Fig. R1. Schematic illustrations and photographs of cooling power and cooling temperature measurement apparatus. a, b) Thermal box apparatus with a feedback-controlled heater for cooling power measurement. c, d) Thermal box apparatus for cooling temperature measurement.

Q2: Figure 4e indicates the solar reflectance of PMMA NP as 0.74, which is rather poor for radiative cooling. Some of commercial white paint can achieve 85%. This is fine since the actual sample is PMMA_{HPA}. However, this is not consistent with the results in Figure 5d that the PMMA_{NP} almost achieved daytime cooling as its temperature was very similar to the ambient temperature. This should not be the case for a coating of 74% solar reflectance. It could be possible that the copper surface helps a lot in the outdoor tests such that the real solar reflectance is higher. Again, the authors should clearly indicate and differentiate what are the optical properties of the film alone and the film plus substrate (copper).

A2: First, during the outdoor cooling temperature measurements for four types of films, the copper plates were removed and the adhesive resistant temperature detectors were directly mounted on the back surface of the films to detect real-time temperature of the samples. (This test method was following one provided by Zhai, Y. *et al. Science* **355**, 1062-1066 (2017))

Second, as the Reviewer said that the solar reflectance of PMMA_{NP} ($\bar{\rho}_{NP} = 0.74$) is rather poor for radiative cooling, but the temperatures of the PMMA_{NP} film just rise to ~ 0.3 °C above the ambient temperature at noon. The possible reasons are as follows:

(1) As shown in Fig. 4e, the PMMA_{NP} film presents a high reflectance level in the UV-Vis wavelengths ($\bar{\rho}_{NP-UV-Vis} = 0.93$), which contains most of the solar radiation. The reflectance in UV wavelengths is even higher than PMMA_{HPA} film. This strong optical scattering of UV-Vis wavelengths gives our PMMA_{NP} surface a white and opaque appearance.

(2) The PMMA_{NP} film has a strong thermal emittance in the LWIR transparency window ($\bar{\epsilon}_{NP} = 0.96$), which is much higher than commercial white paints. This enables strong emission of heat to the cold sink of outer space through the atmospheric transparency window.

(3) Undeniably, the fabricated sample may have some optical errors ($\bar{\rho}_{NP} = 0.74 \pm 0.05$, $\bar{\epsilon}_{NP} = 0.96 \pm 0.01$).

Therefore, the PMMA_{NP} film can achieve a subambient cooling of ~ 6.5 °C during the night but rise to ~ 0.3 °C above the ambient temperature at noon.

Q3: The simulation of the solar reflectance in Figure S7 is lower than the experimental characterization in Figure 2a. Usually the theoretical estimation is similar or higher than experimental results due to imperfection in fabrication process. Could the authors comment on the possible causes?

A3: Thanks for the kind suggestion. Actually, owing to the computational limits and overlong computing time, we simplified the structure models and reduced the effective thickness of the film to ~ 80 μm (~ 160 μm effective thickness was used in experiments in Fig. 2a). Please see the details in Supplementary pages 4-5. Evidently, the simulated solar reflectance in Figure S7 ($\bar{\rho}_{\text{simulated-}80\ \mu\text{m}} \approx 0.90$) is higher than the measured solar reflectance of PMMA_{HPA} film with ~ 80 μm effective thickness in Fig. 3a ($\bar{\rho}_{\text{measured-}80\ \mu\text{m}} \approx 0.88$).

Q4: The authors claim that the previous approaches with silver mirror "appears to be costly". However, the fabrication method in this work involves SiO₂ microspheres with narrow particle size range and HF etching. As such, it is difficult to agree with the authors' claim that their film is cheaper simply based on raw material cost due to the high possible manufacturing process costs. The porous polymer or recent progress of cooling paints can be packaged into cans and sprayed rather conveniently, and silver layer can be rolled easily as well, which could also translate into cheaper installations costs than the current approach.

A4: The previous statement "appears to be costly" in the introduction is not appropriate and we have made corrections according to the Reviewer's comments. Please see the blue words in page 4. In addition, we have added the references 18, 35, 38 and 39 in the introduction, including porous polymer and cooling paints, to better review recent outstanding progress in the radiative cooling field.

We agree with the Reviewer that we cannot claim that the film is cheaper simply based on raw material cost. However, instead of using a reflective metallic mirror or a multilayer photonic structure, our study and the unique structure provide a new prototype and may pave the way for designing and fabrication of high-performance PDRC systems.

REVIEWERS' COMMENTS

Reviewer #1 (Remarks to the Author):

I am satisfied with the changes. I think the paper can be accepted now.